# Mitigating Overconfidence in Out-of-Distribution Detection by Capturing Extreme Activations

**Mohammad Azizmalayeri**[1]      **Ameen Abu-Hanna**[1]      **Giovanni Cinà**[1,2,3]

[1]Department of Medical Informatics, Amsterdam Public Health Research Institute, Amsterdam UMC, University of Amsterdam, The Netherlands
[2]Institute of Logic, Language and Computation, University of Amsterdam, The Netherlands
[3]Pacmed, Amsterdam, The Netherlands

## Abstract

Detecting out-of-distribution (OOD) instances is crucial for the reliable deployment of machine learning models in real-world scenarios. OOD inputs are commonly expected to cause a more uncertain prediction in the primary task; however, there are OOD cases for which the model returns a highly confident prediction. This phenomenon, denoted as "overconfidence", presents a challenge to OOD detection. Specifically, theoretical evidence indicates that overconfidence is an intrinsic property of certain neural network architectures, leading to poor OOD detection. In this work, we address this issue by measuring extreme activation values in the penultimate layer of neural networks and then leverage this proxy of overconfidence to improve on several OOD detection baselines. We test our method on a wide array of experiments spanning synthetic data and real-world data, tabular and image datasets, multiple architectures such as ResNet and Transformer, different training loss functions, and include the scenarios examined in previous theoretical work. Compared to the baselines, our method often grants substantial improvements, with double-digit increases in OOD detection AUC, and it does not damage performance in any scenario.

## 1 INTRODUCTION

Post-deployment, neural networks may encounter out-of-distribution (OOD) samples coming from a distribution different than that of the training set. This may be due to reasons such as data shift, variations in data collection protocol, and input noise, among others [Bandi et al., 2018, Koh et al., 2021, Zadorozhny et al., 2022]. The predictions on such samples can be unreliable, which causes major concerns for deployment in high-stakes applications. A solution to this problem is OOD detection, which is intended to identify OOD inputs in real time before serving any prediction [Yang et al., 2021, Zimmerer et al., 2022].

A common assumption in OOD detection is that machine learning (ML) models are more uncertain about OOD inputs compared to in-distribution (ID) data. This rationale underpins different metrics to identify OOD inputs such as maximum softmax probability (MSP) or entropy [Hendrycks and Gimpel, 2017], which in turn relate to different ways in which one can measure uncertainty. However, for several metrics of uncertainty, it has been demonstrated that ML models can return overconfident predictions on some kinds of OOD inputs, e.g., abnormally high softmax confidences [Nguyen et al., 2015]. This can drastically reduce OOD detection performance.

This phenomenon has been theoretically investigated for feed-forward models with ReLU activation function in the studies by Hein et al. [2019], Ulmer and Cinà [2021]. For OOD instances generated from ID data by scaling a single variable, they prove that the output probability vector can converge to a one-hot vector. As a result, a model employing uncertainty measures like MSP or entropy would be highly overconfident in classifying such OOD cases as ID.

In this work, we address the overconfidence problem in OOD detection methods. For this purpose, we propose to adjust the novelty score—the score assigned to each input to classify it as OOD or ID—by adding a second term responsible for capturing overconfidence. Inspired by the observation that OOD inputs cause outsized activation values in the neural networks [Sun et al., 2021], we suggest defining this term by capturing extreme activation values (CEA). In practice, this involves taking the $\ell_2$-norm of extreme activation values (defined as surpassing a specified threshold) at the penultimate layer of models. If the threshold is chosen appropriately over a validation set, ID activations remain below the cutoff, and the term only captures overconfidence caused by OOD samples. Our method is displayed in Fig. 1.

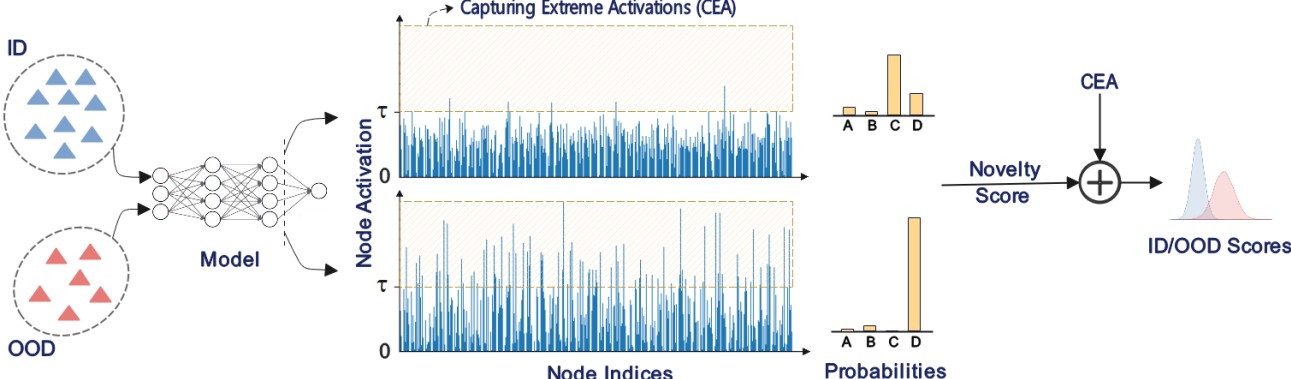

Figure 1: Visual representation of the proposed method. We measure the $\ell_2$-norm of extreme activation values larger than the threshold $\tau$ (CEA) as an indicator of overconfidence caused by OOD samples and add it to the original novelty score computed based on the probabilities and activation values to generate the final novelty scores.

To assess the effectiveness of the proposed method, we adopt the experimental settings outlined in Ulmer and Cinà [2021], as the authors define the context stated earlier involving piece-wise linear models and synthetic OOD, where the impact of overconfidence on OOD detection becomes evident. In addition, we experiment with alternative architectures and OOD data such as tabular ResNet and Transformer and real-world OOD sets, where overconfidence may not occur. Our experiments also include models trained with LogitNorm [Wei et al., 2022], a custom loss mitigating overconfidence. Furthermore, we conduct experiments with image data to evaluate how the results extend to other modalities.

Results demonstrate that many baseline OOD detection methods can benefit from CEA as it remarkably enhances the OOD detection performance of several baselines across different settings. For example, averaged results over 5 different tabular datasets indicate a 45.6% improvement in the AUC of detecting synthesized OODs by MSP. Moreover, MSP performance on average increases by 41.1% in the experiments with a real-world tabular OOD set. Our findings also shed light on the different factors influencing overconfidence such as network architecture and ID data heterogeneity. The advantage of our proposal is that it can be incorporated into any method without requiring any change to the original technique or adding much computational overhead. Consequently, this method can potentially be applied to a variety of settings, improving the reliability of OOD detection methods across the board. All experiments are fully reproducible and the code is provided open access[1].

## 2 PRELIMINARIES AND RELATED WORK

**OOD detection:** For identifying OOD instances, we require a function $f$ assigning larger novelty scores to OOD inputs

[1]https://github.com/mazizmalayeri/CEA.

compared to ID ones and a threshold $\beta$ such that:

$$G(x, f, \beta) = \begin{cases} \text{OOD} & f(x) \geq \beta \\ \text{ID} & f(x) < \beta \end{cases} . \qquad (1)$$

Common choices for $f$ are methods that train a new model to estimate the distribution of ID data such as approaches based on an auto-encoder [Zhou, 2022], and post-hoc detection methods, which are elaborated on below. In this study, the latter are of particular interest since they suffer from the impact of overconfidence in the generated novelty score.

**Post-hoc OOD detection:** Assuming that a model is trained for a certain task (which could be anything, from sentiment classification to mortality prediction), post-hoc methods can be employed to identify OOD inputs without retraining a new model, which makes them an appealing choice in many applications [Yang et al., 2021]. The novelty score is often generated based on the class probabilities or the internal representations of the pre-trained neural network. For example, EBO [Liu et al., 2020] utilizes an energy score instead of a softmax score since it aligns with the probability density of inputs and suffers less from overconfidence, or MDS [Lee et al., 2018] measures the distance of each input from class-conditional Gaussian distributions in the feature space. More examples can be found in Appendix A.

**Overconfidence in OOD detection:** For the remainder of this paper, we take the maximum softmax probability of the predicted classes as the measure of confidence of the model (we employ certainty and confidence as synonyms). With *overconfidence* we refer to the phenomenon of having a level of confidence in the predicted class that increases as we move away from ID data in the feature space. This in turn engenders a decrease in OOD detection AUC as we transition away from ID data, contrary to what we would want. In addition to the softmax scores, this phenomenon can extend to the intermediate layers as demonstrated in Sun et al. [2021]: activation values in the internal layers of a

neural network have a different pattern for OOD data. This can reduce the performance of post-hoc detection methods. Among the methods provided for OOD detection, ReAct [Sun et al., 2021] and LogitNorm [Wei et al., 2022] are specifically designed to mitigate this problem.

ReAct addresses this issue by capping the activation values in the intermediate layers of neural networks at an upper limit, thereby the overconfident values will not affect the final prediction and novelty score. Despite the advantages of this method, we argue that it may lose useful information. Accordingly, in contrast to ReAct, we suggest retaining those values when generating the novelty score, but adjusting the novelty score based on the $\ell_2$-norm of extreme activation values (CEA).

LogitNorm is not an OOD detection method, but a loss function designed to alleviate the overconfidence of neural networks. For this purpose, motivated by the insight that increased logit norm during training with softmax cross-entropy loss induces overconfidence, the authors train the model by enforcing a constant norm on the logit vector. Note that this method requires training a new model with a constrained loss that may impact the optimization of the model for the original task.

**Theoretical vulnerability of OOD detection:** The study by Ulmer and Cinà [2021] gives a theoretical explanation of why OOD detection fails under overconfidence. To achieve this, they utilize a known result that feed-forward neural networks with piece-wise linear activation functions partition the input space into polytopes [Arora et al., 2018]. They then use the fact that these networks are component-wise strictly monotonic on each of their polytopes [Croce and Hein, 2018, Hein et al., 2019]. They also prove that if we scale a single variable in an input with a factor $\alpha$, there exists a value $\delta$ such that $\forall \alpha > \delta$, the output always lies in a specific polytope.

Keeping in mind that we stay within a single polytope for any large $\alpha$ and acknowledging the monotonic nature of the polytopes, it is proved under certain conditions that if we scale a variable from the input by $\alpha \to \infty$, the softmax output of the defined neural network converges to a one-hot vector. Using this finding, in their Theorem 1, they conclude that OOD detection fails if we measure uncertainty with metrics like MSP and entropy, as these methods assign smaller novelty scores to such OOD instances compared to ID ones. Furthermore, it has been empirically observed that this phenomenon occurs at a limited $\alpha$ as well [Azizmalayeri et al., 2023].

# 3 METHOD

In this section, we introduce CEA, a method that addresses the problem of overconfidence by modifying the novelty scores based on the outsized activation values. For this pur-

pose, we describe below how the confidence level can be integrated into the OOD detection setup.

## 3.1 CONSIDERING OVERCONFIDENCE IN NOVELTY SCORE

In order to deal with overconfidence in OOD detection, we suggest directly taking it into account as part of the novelty score generation. We propose adding a new term to the novelty score which is non-zero only when an OOD input causes overly confident prediction; otherwise, the new term remains close to zero and the original novelty score would be retained. In summary, we propose to change Eq. 1 to:

$$G(x, f, g, \beta) = \begin{cases} \text{OOD} & (f(x) + \lambda\, g(x)) \geq \beta \\ \text{ID} & (f(x) + \lambda\, g(x)) < \beta \end{cases}, \quad (2)$$

where $\beta$ is a threshold for classifying a sample as OOD, $f(x)$ returns the novelty score, $g(x)$ is the new term responsible for indicating overconfidence, and $\lambda$ controls the tradeoff between $f$ and $g$. The function $g(x)$ should have the following characteristics:

- The value returned by $g(x)$ for ID data should be smaller or equal to the value returned by $g(x)$ for OOD data.
- The value returned by $g(x)$ should monotonically increase as the overconfidence level rises, e.g., when amplifying the scaling factor $\alpha$ for synthesizing the OOD instances.

The first condition guarantees that the addition of $g(x)$ will not adversely impact the performance of the original novelty score $f(x)$ for OOD detection, and the second one is directed towards the primary objective of introducing g(x), namely highlighting the existence of overconfidence. Note that $g(x)$ alone may not be sufficient for OOD detection as OOD instances without overconfidence will not be spotted. One may also think of using $g(x)$ as a trigger for $f(x)$, meaning that the latter is used only when the former does not trigger. This approach however requires an additional hyperparameter to decide when $g$ would raise a flag.

## 3.2 OVERCONFIDENCE MEASURE

In this section, we present a choice for $g(x)$ that meets the specified conditions and can be applied to any architecture. It has been observed that OOD data can lead to activation patterns in neural networks that are significantly different from ID data, i.e., activation units with extremely large values, which results in overconfident predictions [Sun et al., 2021]. We demonstrate in the subsequent theorem that one kind of overconfident behavior on the side of the model entails the presence of extreme activations in the penultimate layer.

**Theorem 1.** *Let $x \in R^D$ and suppose $\alpha$ is a scaling vector. Now $x' = \alpha \odot x$ can be considered as an OOD example if*

**Algorithm 1** Simple code for the proposed method.

---

**Input:** Prediction model $k_\theta$, sample $x$, OOD detection method $f$.

**Parameters:** Coefficient $\lambda$, Threshold $\tau$.

$x_{activations}, x_{logits} = k_\theta(x)$  ▷ Activations in penultimate layer.
$NS = f(x_{activations}, x_{logits})$  ▷ Original novelty score ($f(x)$).
$CEA = max(x_{activation} - \tau, 0)$  ▷ Capturing extreme activations.
$CEA = \|CEA\|_2$  ▷ $\ell_2$-norm of extreme values ($g(x)$).
$NS = NS + \lambda\, CEA$  ▷ Modifying $NS$ based on $CEA$.

**Output:** Modified novelty score $NS$.

---

$\alpha$ *is large enough. Let* $h_\theta$ *be any neural network whose last layer is linear, generating an overconfident prediction for class* $c$ *on* $x'$ *as:*

$$\lim_{\alpha_d \to \infty} \sigma(h_\theta(x'))_c = 1, \qquad (3)$$

*where* $\sigma$ *is the Softmax function. Then, we infer that there exists at least a dimension in which the output of the penultimate layer goes to infinity in the limit.*

*Proof.* The proof is available in Appendix C. $\qquad\square$

This finding suggests that extreme activations in the penultimate layer can be an indicator of overconfidence in the prediction. Hence, we can measure the magnitude of extreme activations, denoted as CEA, as a proxy for $g(x)$. To this end, we use the $\ell_2$ norm of node activation values at the penultimate layer of the neural network that are higher than a specified threshold. Accordingly, assuming that $k_\theta(x)$ is the activation vector before the classification layer generated by the prediction model $k$ with parameters $\theta$ for the input $x$, we define CEA as:

$$CEA(x, k_\theta, \tau) = \|max(k_\theta(x) - \tau, 0)\|_2, \qquad (4)$$

where $\tau$ is the specified threshold. We utilize $\ell_2$-norm in our method, but it can potentially be substituted with other norms as well. The pseudocode for computing CEA as a proxy of $g(x)$ and adding it to the original novelty score is provided in Algorithm 1. This selection for $g(x)$ intuitively yields larger values for the overconfident OOD inputs than other OODs as they lead to more outsized activation nodes. Also, with a suitable choice of $\tau$, the values returned for ID data would be comparatively smaller. Therefore, by appropriately selecting hyperparameters, the CEA algorithm can fulfill the specified conditions.

**Hyper-parameter selection:** $\tau$ and $\lambda$ play an important role in the proposed method. We select threshold $\tau$ such that it remains above the feature values of ID data. Hence, $g(x)$ is close to zero for ID data, while it can capture outsized feature values in OOD instances. To tune these values, we use a validation set from ID data $\mathcal{D}_{val}$ to extract their activation values at the penultimate layer of the prediction model. $\tau$ can be set to the maximum activation value extracted from the validation data; however, in presence of outliers, such a choice might lead to a very large $\tau$ that does not let $g(x)$ capture the overconfidence even in OOD cases. Alternatively, we use the activation value at the $p$-th percentile to avoid noisy activation values. In our study, we set $p = 99.9$ for tabular datasets and $p = 99.999$ for images. Furthermore, we also scale this value by a factor of $\rho = 1.1$ to ensure that most ID feature values remain below the threshold.

The coefficient $\lambda$ is determined based on the average $f(x)$ and $g(x)$ over $\mathcal{D}_{val}$. More specifically, $\lambda$ is computed as:

$$\lambda = \gamma \, \Big| \frac{\sum_{x \in \mathcal{D}_{val}} f(x)}{\sum_{x \in \mathcal{D}_{val}} g(x)} \Big|, \qquad (5)$$

where $\gamma$ lets us control the tradeoff between $f(x)$ and $g(x)$. In our study, we set $\gamma = 1$. We conduct an ablation study on the parameters $p$ and $\gamma$ in the experiments.

It should be noted that we only use a validation from ID data to find the hyper-parameters. Still, we could tune $\tau$ and $\lambda$ further using a diverse set of OOD examples. This involves assessing the OOD detection performance across different ranges of these parameters. However, this requires a set of diverse OOD data which covers different kinds of OOD examples that model may face in practice, which is not available in many cases.

### 3.3 EXPERIMENTAL SETUP

To investigate the effectiveness of our proposed method in contrast to baseline OOD detection methods, we follow the theoretical findings described earlier, where the neural network has a provably higher level of confidence for some kinds of OOD cases compared to IDs. Accordingly, we incorporate the assumptions used in Theorem 1 of Ulmer and Cinà [2021] as follows:

- A ReLU classifier constituted of fully connected layers with ReLU activation function is considered as the basic architecture of the prediction model. This architecture is a piece-wise affine function that leads to overconfident predictions as discussed.
- OOD instances are generated by scaling a single input variable from the ID data by an $\alpha$ factor. We repeat synthesizing OOD versions 50 times with different variables and average the results over them to minimize the effect of the selected variable. If a dataset has a smaller number of variables, we use each of them once. Moreover, we use different values of $\alpha$ to see how that affects the results. We are interested in such OODs since for large $\alpha$ they lead to overconfident predictions according to the theory.

The experiments proceed as follows. First, a prediction model is trained on a specific dataset. Then, post-hoc OOD detection methods with and without our method are applied

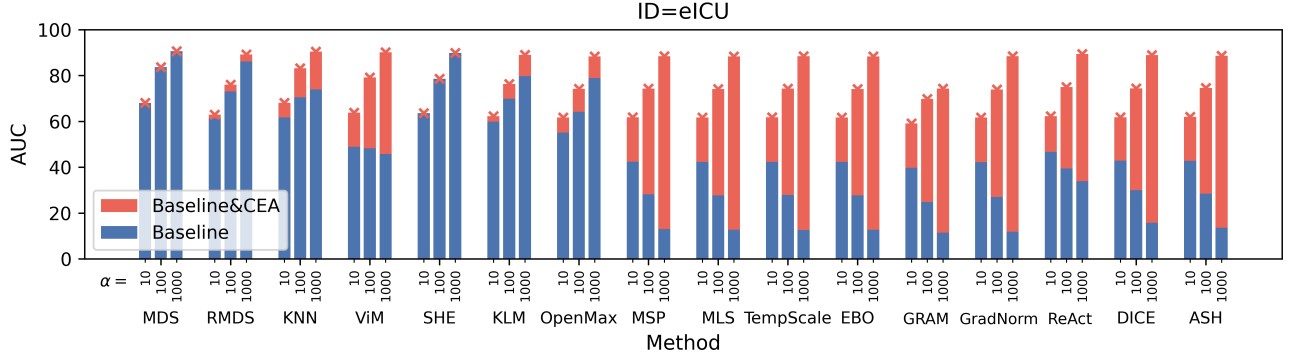

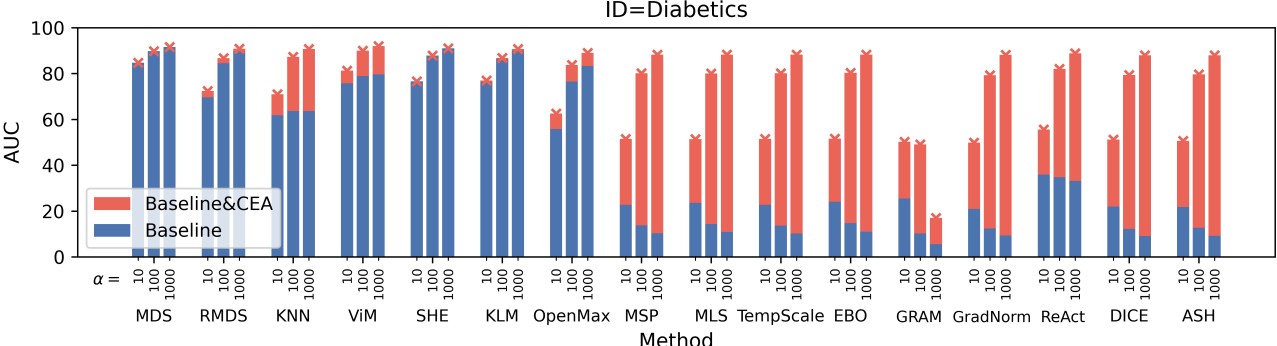

Figure 2: OOD detection performance with and without CEA using the eICU (top) or Diabetics (bottom) datasets as ID and synthesized OOD data obtained by scaling. The blue bars are positioned in front of the red ones and cross markers are employed to emphasize the top of the red bars. The scaling factors $\alpha$ and baselines are presented under each bar.

to the prediction model to examine whether they can discriminate OOD instances from IDs and the extent to which they suffer from overconfident predictions. AUC is used as the discrimination criterion and the results are averaged over three repetitions of experiments with different randomization seeds to increase reliability. We investigate more complicated architectures and some other OOD groups in the experiments as well, to see how the results extend to those scenarios. In the following, we provide details about these architectures and OODs along with the datasets and OOD detection baselines used in the experiments.

**Tabular datasets:** We consider 5 different tabular datasets consisting of eICU [Pollard et al., 2018], MIMIC-IV [Johnson et al., 2023], Diabetic Retinopathy Debrecen [Antal and Hajdu, 2014], Dry Bean [Koklu and Ozkan, 2020], and Wine Quality [Cortez et al., 2009]. These datasets are drawn from different domains and aim at diverse prediction tasks such as mortality prediction or wine type. In addition, they include large-scale and unbalanced (e.g., eICU) datasets. Detailed information about the datasets can be found in Appendix B. For the first three datasets, we use $\alpha \in [10, 100, 1000]$ in synthesizing the OOD instances, while opting for $\alpha \in [2, 3, 4]$ for the remaining ones as they exhibit overconfident behavior at a smaller scale.

**Real-world tabular data as OOD:** Besides synthesized OOD groups, it is valuable to examine real-world OOD instances as well. Among the tabular datasets mentioned above, eICU and MIMIC-IV datasets share a lot of similar variables. Consequently, we can employ each of them as the OOD data for the other one. For this purpose, we only use the shared variables in these two datasets. This experiment is also interesting in that it explores the effect of data homogeneity on overconfidence: as opposed to eICU, which comprises the data of several hospitals, MIMIC-IV is more homogeneous since it sources data from a single center.

**Architectures:** In addition to the ReLU classifier, we also employ tabular ResNet and Transformer [Gorishniy et al., 2021] in the experiments. It has been empirically observed that tabular Transformers can mitigate the overconfidence phenomenon [Azizmalayeri et al., 2023], which allows us to evaluate our method in combination with a prediction model that suffers from this issue only marginally.

**Image datasets:** In addition to tabular datasets, we also consider three widely used image datasets MNIST [Deng, 2012], CIFAR-10, CIFAR-100 [Krizhevsky et al., 2009]. We train a ReLU classifier and a ResNet-32 model on each of these datasets. Within images, we only set $\alpha = 10$ for the synthesized OODs since it is enough to show overcon-

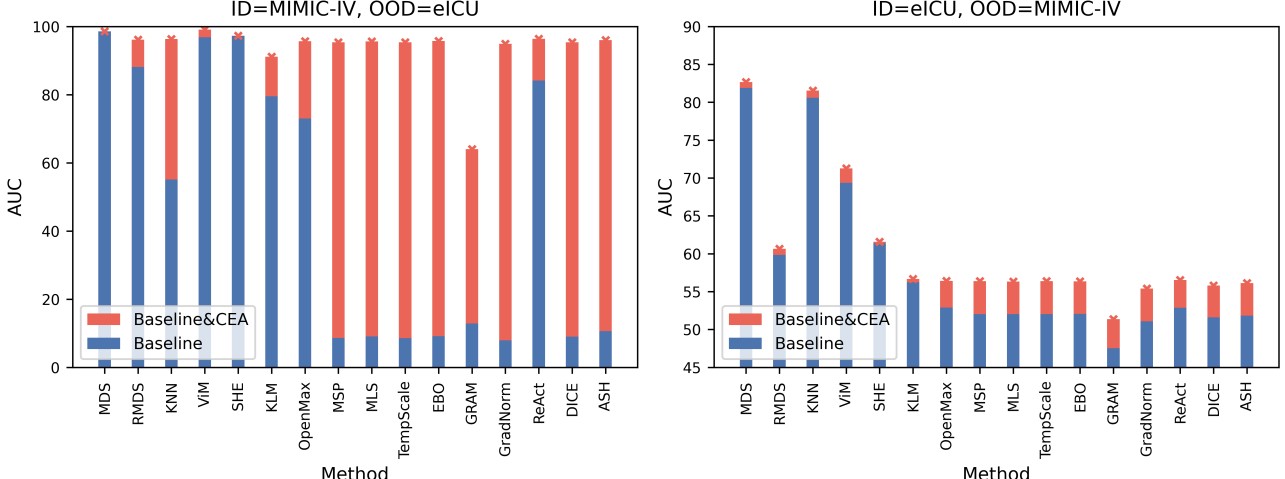

Figure 3: OOD detection performance with and without CEA using MIMIC-IV as ID and eICU as OOD (left) and the other way around (right). The blue bars are positioned in front of the red ones and cross markers are employed to emphasize the top of the red bars.

fidence, and also use adversarial examples generated by maximizing the cross-entropy loss of the model itself via a PGD-20 attack with $\epsilon = 32/255$ as another way of generating overconfident OOD instances [Nguyen et al., 2015, Madry et al., 2018]. Furthermore, we also report the average detection performance on real-world OOD datasets, namely Fashion MNIST [Xiao et al., 2017], MNIST, SVHN [Netzer et al., 2011], CIFAR-10, and CIFAR-100 (the one used as ID is excluded in the averaging).

**Baseline OOD detection methods:** A wide range of post-hoc OOD detection baselines are considered in this study following recent benchmarks in OOD detection [Yang et al., 2022, Zhang et al., 2023, Azizmalayeri et al., 2023]. These baselines include both commonly used and top-performing post-hoc OOD detection methods. More information about these methods can be found in Appendix A. In addition, we also consider the same baselines applied to an MLP architecture with LogitNorm [Wei et al., 2022].

## 4 RESULTS

**Evaluation with synthetic OOD** The results of the experiments on tabular datasets are displayed in Fig. 2 for eICU and Diabetics datasets, and in Appendix D for the rest. Based on the results, our method gives a net positive improvement on most baselines, with two exceptions that are not affected, MDS and SHE. Furthermore, the improvements are more notable in detection methods such as MSP and EBO which rely on the probabilities to generate the novelty scores.

In addition, it is expected that OOD data generated with a larger scaling factor are detected better as they are farther

away from the ID data. However, certain baselines present a different behavior, and their performance is decreased by increasing the scaling factor. This is a sign of overconfidence in the prediction model as the detection method assigns a lower novelty score to the OOD instances farther from the ID data. As hoped, our method addresses this issue: combining the baselines with our method leads to consistently better performance for larger scaling factors.

**Real-world tabular data as OOD** To assess the performance on real-world tabular OOD datasets, we consider MIMIC-IV as OOD set for eICU and vice versa. Results of this experiment are illustrated in Fig. 3. Similarly to synthesized OODs, our method significantly improves the OOD detection performance across several baselines without negatively affecting any of them.

The results also indicate that our method grants more improvement when the model is trained on MIMIC-IV as ID. This shows that the prediction model trained on MIMIC-IV suffers more from overconfidence compared to the eICU dataset (see Discussion). Finally, it is worth noting that the addition of our methods improves all OOD detection performances above the chance threshold of 0.5 AUC (often far better) 'reversing' the effect of overconfidence.

**Other architectures** The architecture of the prediction model plays an important role in its overconfidence. Thus, we employ tabular ResNet and Transformer [Gorishniy et al., 2021] to evaluate our approach. Table 1 displays the results, demonstrating that the addition of CEA outperforms numerous baseline detection methods when applied to ResNet (due to space limitation, we show only some baselines and put the rest in Appendix D). However, the improvements

Table 1: AUC of OOD detection with and without CEA using tabular ResNet and Transformer as the prediction model. We use eICU and Diabetics as ID and synthesize the OOD data by scaling factor $\alpha$. Superior results are emphasized in bold unless the two are equal.

| ID | Method | ResNet | | | Transformer | | |
| | | $\alpha = 10$ | $\alpha = 100$ | $\alpha = 1000$ | $\alpha = 10$ | $\alpha = 100$ | $\alpha = 1000$ |
| | | Baseline / Baseline&CEA | | | | | |
|---|---|---|---|---|---|---|---|
| eICU | MDS | **77.8** / 77.6 | 91.7 / **91.8** | 93.6 / 93.6 | 63.3 / 63.3 | 83.6 / 83.6 | 90.7 / 90.7 |
| | KNN | 72.0 / **74.7** | 86.8 / **90.5** | 89.5 / **93.3** | 59.9 / 59.9 | 79.5 / 79.5 | 90.1 / 90.1 |
| | ViM | 75.4 / 75.4 | 91.4 / 91.4 | 93.7 / 93.7 | 60.0 / 60.0 | 80.5 / 80.5 | 90.3 / 90.3 |
| | MSP | 47.9 / **69.5** | 30.6 / **87.3** | 13.2 / **93.6** | 51.7 / **52.5** | 56.1 / **58.3** | 71.7 / **73.5** |
| | EBO | 46.4 / **69.2** | 28.6 / **87.1** | 13.2 / **93.6** | 51.6 / **52.3** | 56.1 / **57.9** | 71.4 / **73.0** |
| | ReAct | 61.6 / **70.3** | 71.8 / **88.0** | 76.1 /**93.6** | 51.9 / **52.5** | 56.6 / **58.3** | 72.0 / **73.7** |
| | Gram | 35.4 / **55.0** | 16.0 / **44.8** | 6.8 / **24.0** | 50.8 / **51.7** | 54.4 / **57.0** | 68.3 / **69.9** |
| Diabetics | MDS | **85.9** / 85.8 | 90.2 / **90.3** | 91.8 / 91.8 | 84.3 / 84.3 | 89.5 / 89.5 | 91.3 / 91.3 |
| | KNN | 80.0 / **83.9** | 85.0 / **89.7** | 87.0 / **91.9** | 80.3 / 80.3 | 88.3 / 88.3 | 90.8 / 90.8 |
| | ViM | 79.2 / **84.9** | 83.9 /**90.8** | 85.5 / **92.5** | 84.2 / 84.2 | 90.6 / 90.6 | 92.2 / 92.2 |
| | MSP | 25.6 / **67.4** | 15.5 / **86.3** | 10.5 / **90.3** | 38.4 / **39.9** | 47.3 / **48.2** | 61.8 / **62.7** |
| | EBO | 35.1 / **68.4** | 23.0 / **86.1** | 18.5 / **90.2** | 43.0 / **44.0** | 50.8 / **51.4** | 66.8 / **68.1** |
| | ReAct | 44.3 / **69.9** | 43.4 / **86.6** | 42.8 / **90.4** | 45.8 / **46.9** | 52.2 / **52.9** | 66.6 / **68.1** |
| | Gram | 21.5 / **65.2** | 11.9 / **74.8** | 6.7 / **38.8** | 50.6 / **52.1** | 54.2 / **54.7** | 64.3 / **65.1** |

Table 2: AUC of OOD detection with and without CEA on the model trained with the LogitNorm loss. We use eICU and Diabetics as ID and synthesize the OOD data by scaling factor $\alpha$. Superior results are emphasized in bold unless the two are equal.

| ID | $\alpha$ | MDS | KNN | ViM | MSP | EBO | ReAct | Gram |
| | | Baseline / Baseline&CEA | | | | | | |
|---|---|---|---|---|---|---|---|---|
| eICU | 10 | **72.8** / 72.6 | 57.7 / **68.2** | 45.9 / **59.2** | 55.0 / **67.3** | 55.0 / **67.3** | 53.5 / **65.8** | 37.1 / **60.1** |
| | 100 | 85.6 / **85.7** | 63.9 / **82.4** | 43.6 / **75.4** | 61.7 / **82.7** | 61.7 / **82.7** | 55.1 / **80.4** | 21.7 / **62.6** |
| | 1000 | 90.5 / 90.5 | 66.1 / **89.2** | 42.6 / **87.2** | 64.6 / **89.9** | 64.7 / **90.0** | 54.2 / **89.2** | 11.3 / **42.7** |
| Diabetics | 10 | 84.7 / 84.7 | 82.9 / **83.4** | 84.2 / 84.2 | 35.2 / **65.0** | 35.2 / **65.0** | 20.3 / **61.0** | 23.4 / **51.6** |
| | 100 | 89.6 / 89.6 | 88.3 / **89.1** | 89.7 / 89.7 | 32.6 / **85.7** | 32.6 / **85.7** | 11.9 / **84.2** | 12.5 / **55.0** |
| | 1000 | 91.7 / 91.7 | 90.5 / **91.3** | 91.9 / 91.9 | 31.7 / **90.0** | 31.9 / **90.1** | 9.1 / **89.1** | 9.4 / **29.2** |

with the Transformer are marginal. This aligns with prior observations that Transformer as an architecture mitigates the effect of overconfidence [Azizmalayeri et al., 2023].

It is noteworthy that the OOD detection performance of the pure baseline methods is better on average on the Transformer model as it internally addresses the overconfidence. However, results on ResNet plus CEA often get better than Transformer with the same advantage (especially for MSP, EBO, and React). Hence, while changing the architecture of the prediction model itself can be a solution to overconfidence, its capability for OOD detection is still highly dependent on the way OODs are singled out.

**LogitNorm training** To assess the impact of LogitNorm training, we also train prediction models with this loss instead of softmax cross-entropy loss. Results are provided in

Table 2. According to this table, our method still manages to improve on the models trained with this dedicated loss across different datasets and baselines.

Comparing the results from this table and Figure 2 indicates that LogitNorm itself leads to better OOD detection as expected. Nevertheless, it does not eliminate overconfidence in the prediction model, e.g., OOD detection using LogitNorm plus MSP (or EBO, or ReAct, or Gram) on Diabetics results in worse performance than a random binary classifier.

**Extension to images** In this section, we evaluate the OOD detection performance within the image domain. Results for MNIST and CIFAR-10 datasets are presented in Table 3 and additional results for CIFAR-100 can be found in Appendix D. For synthesized OOD sets, our method significantly improves the performance of many baselines regardless of the

Table 3: AUC of OOD detection with and without CEA in image datasets. MNIST and CIFAR-10 serve as ID, and OOD sets are synthesized by i) scaling or ii) an adversarial attack, or iii) selected from other datasets. ResNet-32 and ReLU MLP classifiers are used as the prediction model. Superior results are in bold unless the two are equal.

| | | ReLU MLP | | | ResNet-32 | | |
|---|---|---|---|---|---|---|---|
| | | Scale | Attack | Other | Scale | Attack | Other |
| ID | Method | Baseline / Baseline&CEA | | | | | |
| MNIST | MDS | 64.2 / **64.3** | **98.5** / 98.1 | 88.7 / **90.2** | 59.5 / 59.5 | 99.9 / 99.9 | 99.9 / 99.9 |
| | KNN | 62.1 / **62.4** | 73.1 / **84.6** | 97.6 / **98.2** | 54.6 / 54.6 | 99.2 / **99.7** | 99.9 / 99.9 |
| | ViM | 60.2 / 60.2 | 67.4 / **68.0** | 98.0 / 98.0 | 58.3 / 58.3 | 99.9 / 99.9 | 99.9 / 99.9 |
| | MSP | 46.3 / **63.2** | 26.1 / **73.1** | 77.5 / **89.9** | 52.5 / **54.3** | 59.7 / **97.3** | 98.3 / **98.7** |
| | EBO | 41.3 / **61.3** | 7.0 / **44.2** | 74.7 / **92.3** | 47.5 / **67.7** | 11.7 / **92.8** | 95.5 / **97.0** |
| | ReAct | 56.4 / **60.3** | 28.7 / **67.7** | 88.8 / **93.6** | 60.9 / **61.0** | 86.7 / **97.8** | 98.6 / **98.8** |
| | Gram | 34.1 / **48.9** | 2.8 / **12.8** | 26.8 / **36.4** | 43.8 / **44.8** | 1.6 / **71.2** | 53.0 / **62.0** |
| CIFAR-10 | MDS | 97.9 / 97.9 | 95.4 / **95.5** | **61.8** / 61.7 | 99.8 / 99.8 | 98.0 / **99.1** | 31.1 / 31.0 |
| | KNN | 77.7 / **97.4** | 65.3 / **80.4** | 54.5 / **56.4** | 99.5 / **99.6** | 10.0 / **71.1** | 86.9 / **87.0** |
| | ViM | 71.2 / **95.2** | 43.3 / **50.5** | 63.5 / **64.3** | 70.6 / **95.6** | 0.0 / **58.6** | 90.4 / 90.4 |
| | MSP | 14.3 / **90.0** | 11.0 / **39.0** | 57.3 / **59.1** | 88.0 / **99.3** | 0.2 / **74.4** | 88.4 / **88.7** |
| | EBO | 3.9 / **4.4** | 5.9 / **6.3** | 50.0 / 50.0 | 78.0 / **96.7** | 0.0 / **59.0** | 90.4 / 90.4 |
| | ReAct | 71.6 / **94.4** | 72.2 / **81.3** | 55.5 / **57.7** | 97.5 / **99.0** | 13.0 / **73.4** | 81.8 / **82.0** |
| | Gram | 22.3 / **94.1** | 8.2 / **25.9** | 61.6 / **63.1** | 13.8 / 13.8 | 0.0 / **1.6** | 70.4 / 70.4 |

choice of prediction model and dataset. Furthermore, it is similarly effective with real-world OOD sets when applied to the ReLU MLP. Nevertheless, the improvements become marginal with the ResNet architecture, to the extent that on heterogeneous ID data such as CIFAR-100 results with and without CEA are the same for real-world OOD sets.

**Ablation study on hyperparameters**  $\tau$ and $\lambda$ are the main hyperparameters in our method, regulated via $p$ and $\gamma$, respectively. In this section, we examine the effects of these parameters. To achieve this goal, we evaluate the OOD detection performance on the Diabetics dataset across various values of $p$ and $\gamma$ in Fig. 4. This figure demonstrates that the proposed method can improve the OOD detection for a wide range of choices for these parameters. Nevertheless, it is advisable to fine-tune these parameters to achieve optimal results. Note that we have used the fixed set of parameters described in section 3.2. Therefore, the outcomes of our method could potentially be enhanced by identifying the optimal hyperparameters for each detection method and dataset.

Based on Fig. 4, a higher value of $p$ guarantees that CEA only positively influences the performance of baseline OOD detection methods. However, it might reduce the capability to detect overconfident OOD instances, as it raises the threshold $\tau$.

For investigating $\gamma$, we have set $p$ such that it results in a threshold that remains above node activation values of ID data. Consequently, our method only captures the overconfidence in OOD instances, and increasing $\gamma$ results in a better

performance. However, we do not recommend using a large $\gamma$ if the threshold has not been set carefully. In addition, note that a small $\gamma$ can also result in neglecting the effect of CEA on the final novelty score.

## 5  DISCUSSION

As a solution to the overconfidence issue in OOD detection, we proposed CEA, a way to adjust the novelty scores of the post-hoc OOD detection methods by adding a new term to the original novelty score. CEA is only activated when an OOD input results in an outsized activation compared to the corresponding values over the ID validation set.

To demonstrate the effectiveness of CEA, we conducted experiments on 16 different baseline OOD detection methods across 5 different tabular datasets spanning a wide range of domains in a context where it has been theoretically verified that overconfidence hurts OOD detection. We also explored alternative settings with real-world OOD sets, other architectures like tabular ResNet and Transformer, and image datasets. There was a significant improvement in the performance of numerous baselines across these settings; however, there were also methods and settings that were not affected much, which are discussed below.

Our method enhances baseline detection methods relying on the class probabilities to generate novelty scores such as MSP, EBO, and DICE more than those that measure a distance in the feature space such as MDS, SHE, and KLM. This is because distance-based methods inherently handle

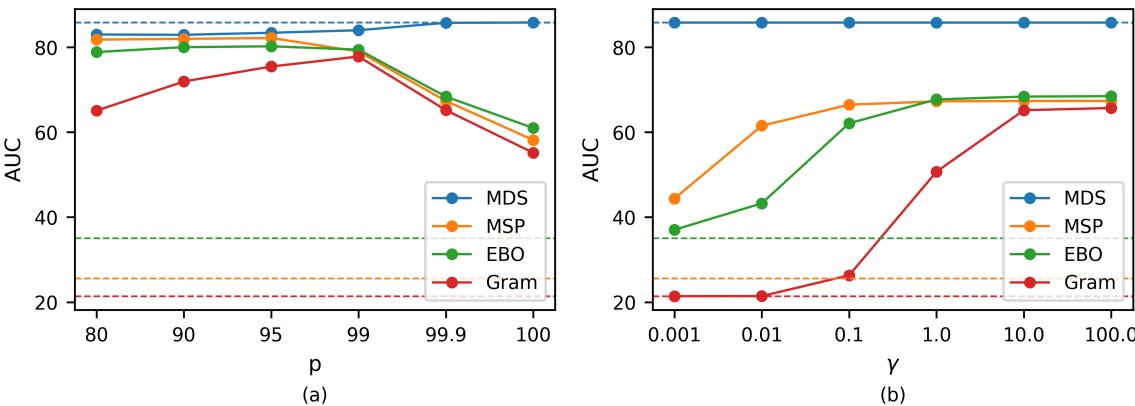

Figure 4: Impact of parameters on the performance of CEA applied on different baseline OOD detection methods within the Diabetics dataset. (a) $\gamma = 10$ and $p$ is changed. (b) $p = 99.9$ and $\gamma$ is changed. The dashed lines indicate the performance of OOD detection methods without CEA ($\gamma = 0$).

overconfidence by relying on distance (measured among internal representations) instead of confidence. Consequently, CEA may not improve much the performance of these kinds of detection methods, as can be observed for MDS in the results. One may be tempted to resort to these approaches instead, but it should be noted that we cannot solely rely on distance-based detection methods as they may not perform well in general, see e.g. MDS applied on the ResNet-32 model trained on the CIFAR-10 dataset. More specifically, another baseline combined with CEA may perform better than methods like MDS. Hence, CEA allows us to replace methods like MDS with other baselines while keeping the benefits of those methods.

Within the architectures evaluated in our study, the Transformer seems to be more robust against overconfidence. This behavior can be explained based on the theoretical understanding of the problem. The proofs provided on overconfidence assume that the model is a piece-wise affine classifier [Hein et al., 2019]. Nevertheless, the Transformer utilizes activation functions and attention mechanism [Vaswani et al., 2017] which are non-linear, violating this assumption. This property results in better OOD detection performance within detection methods such as MSP and EBO; still, when overconfidence is addressed by our proposed method or methods such as MDS, we see that the non-linearity of Transformer is not so beneficial for OOD detection anymore.

The results also showcase that more heterogeneous ID data reduces overconfidence. For example, models trained on eICU and CIFAR-100 are not overconfident in real-world OOD sets as much as models trained on MIMIC-IV and CIFAR-10, respectively. This may explain the observation in the OpenOOD and other benchmarks [Yang et al., 2022, Azizmalayeri et al., 2023] that OOD detection performance is better in some models trained on complex datasets. We also note that model calibration may be another way to

mitigate overconfidence; however, our results in Appendix E demonstrate that calibration improves the OOD detection AUC only marginally.

Lastly, note that the proposed method can be seamlessly incorporated as an extension to any post-hoc OOD detection method, without much computational overhead but with potentially big gains in real-time OOD detection performance. Additionally, it is compatible with other methods proposed for overconfidence such as LogitNorm and ReAct. This property makes this method suitable for many applications, especially those with a high risk of data shift and safety-critical consequences. Even though the applicability of our method may seem impaired by the need to choose hyperparameters, the ablation study demonstrates that the conclusions about CEA are robust within reasonable hyperparameter ranges.

In summary, we believe that the proposed method can increase the reliability of OOD detection methods and benefit a wide range of domains that currently use ML models and OOD detection such as healthcare (e.g., disease recognition or mortality prediction), financial services (e.g., fraud detection), transportation (e.g., autonomous vehicles), and cybersecurity (e.g., identification of OOD network patterns). Our study not only offers a practical solution but also provides insights that open the door to research exploring alternative solutions to overconfidence in OOD detection. On the experimental side, future work can also consider the application of CEA within alternative domains, including but not limited to time-series and text data, to enrich the understanding of the problem. On the theoretical realm, it might be worth investigating which properties of CEA (or the term $g$ more generally) are sufficient to guarantee the absence of overconfidence in OOD detection.

## Acknowledgements

This study was part of the project *Research in clinical prediction models and natural language processing with deep learning* (project number NWO-2021.024) of the research program Computing Time on National Computer Facilities. The computational resources used were financed by the Dutch Research Council (NWO).

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

# Mitigating Overconfidence in Out-of-Distribution Detection by Capturing Extreme Activations
# (Supplementary Material)

**Mohammad Azizmalayeri**[1]    **Ameen Abu-Hanna**[1]    **Giovanni Cinà**[1,2,3]

[1]Department of Medical Informatics, Amsterdam Public Health Research Institute, Amsterdam UMC, University of Amsterdam, The Netherlands
[2]Institute of Logic, Language and Computation, University of Amsterdam, The Netherlands
[3]Pacmed, Amsterdam, The Netherlands

## A    BASELINE POST-HOC OOD DETECTION METHODS

In this section, we provide a summary of baseline OOD detection methods included in our experiments. We have selected these methods following the recent related benchmarks [Yang et al., 2022, Zhang et al., 2023, Azizmalayeri et al., 2023], and detailed information about them can be found in the code and original studies.

**MDS** [Lee et al., 2018]: This method fits a class-conditional Gaussian distribution $\mathcal{N}(\mu_k, \Sigma)$ to the feature vector before the logits. The covariance matrix $\Sigma$ is shared between the classes, but the mean $\mu_k$ is computed separately for each class $k \in \{1, 2, ..., K\}$. The novelty score for an input x with pre-logit features $l_x$ is computed as:

$$\min_k \ (h_x - \mu_k)^T \Sigma (h_x - \mu_k). \tag{6}$$

**RMDS** [Ren et al., 2021]: Motivated by the observation that MDS does not work well on near-OOD data, they suggest fitting a single distribution $\mathcal{N}(\mu, \Sigma)$ to the feature vector before the logits and normalizing the distances measured by MDS as:

$$MDS(h_x) - (h_x - \mu)^T \Sigma (h_x - \mu). \tag{7}$$

They believe that this fix to MDS makes it more robust on near-OOD sets.

**KNN** [Sun et al., 2022]: They suggest non-parametric nearest-neighbor distance for OOD detection. More specifically, they compute the distance to the $k_{th}$ nearest neighbor distance from the training set as the novelty score. The distance is computed based on the embedding extracted from each input.

**ViM** [Wang et al., 2022]: This study suggests that softmax probability and features should be used simultaneously to be capable of detecting different types of OOD. Accordingly, they combine a class-agnostic score from the feature space with the class-dependent scores from logits. More specifically, their method is based on the idea that some information from feature space is not carried to the logits. They recover this information from the principal subspace of features and add it to the score from logits.

**SHE** [Zhang et al., 2022]: This method shares similarities with MDS regarding the intuition behind the method. SHE stores a single class-dependent pattern from the penultimate layer of the neural network over the training set. Afterward, it leverages an energy function defined in Modern Hopfield Network [Ramsauer et al., 2020] to measure the distance between a new input pattern and stored patterns.

**KLM** [Hendrycks et al., 2022]: For each class of data, they average the probability vector extracted from the validation samples classified in the corresponding class by the prediction model. The novelty score for an input is then computed based on the minimum KL distance of its probability vector from the class-dependent probabilities computed earlier.

**OpenMax** [Bendale and Boult, 2016]: On a dataset with $k$ classes, they propose to change the softmax probability such that it generates a probability vector for $k + 1$ classes, where the last class corresponds to the open-set class. To achieve this, they reweight the original probability vector by fitting a Weibull distribution to the class-dependent probabilities.

**MSP** [Hendrycks and Gimpel, 2017]: It is a simple but effective baseline proposed for OOD detection. It is motivated by the intuition that the maximum softmax value for an OOD input should not be as large as ID data. So, it utilizes maximum softmax probability to compute to novelty score.

**MLS** [Hendrycks et al., 2022]: As an alternative to maximum softmax probability, MLS suggest to use maximum logit. Experiments on MLS have demonstrated that it performs better than MSP in large-scale multi-class, multi-label, and segmentation tasks.

**TempScaling** [Guo et al., 2017]: This method calibrates the softmax temperature over a validation before applying MSP for OOD detection.

**EBO** [Liu et al., 2020]: The intuition behind EBO is that $p(y|x)$ used in methods such as MSP should be replaced with $p(x)$, which shows better whether an input $x$ comes from the training distribution. For this purpose, they propose an energy-based framework for OOD detection that computes novelty scores based on the energy score.

**GRAM** [Sastry and Oore, 2020]: They characterize the intermediate representations of the neural network by GRAM matrices. OOD inputs are identified by comparing the values in the GRAM matrices to their respective range computed over the training set.

**GradNorm** [Huang et al., 2021]: The key idea in GradNorm is that the magnitude of gradient back-propagated from the KL distance between the softmax vector and a uniform probability vector would be larger for ID data than that of OOD data. This makes sense as OOD data are generally expected to yield a uniform probability vector for OOD data.

**ReAct** [Sun et al., 2021]: This method rectifies activation units at the penultimate layer of the neural network at an upper limit computed over a validation set. This helps to reduce the impact of overconfidence in the generated novelty score. They suggest applying EBO after rectification, but it can be combined with other detection methods as well.

**DICE** [Sun and Li, 2022]: The idea of DICE is that reliance of neural networks on unimportant weights and units can reduce the OOD detection performance. Accordingly, DICE proposes to rank weights based on a contribution measure and only use the more contributing ones in OOD detection. A simple example of the contribution measure is averaging the output of each weight over a validation set.

**ASH** [Djurisic et al., 2023]: This study extends the neural network sparsification idea and proposes to remove a large proportion of an input's activations and lightly adjust the rest. The change in the weights is case-specific and does not require any statistic from the training set.

## B  DATASET AND TASK DETAILS

In this section, we present information about the datasets and the associated prediction tasks for which they are employed. These datasets are publicly available (some need access authorization).

### B.1  TABULAR

**eICU:** The eICU Collaborative Research Database is a dataset containing health data from the patients admitted to the United States ICUs in 2014-2015. This dataset can be accessed through PhysioNet [1] but requires to be a credentialed user on the website. For pre-processing this dataset, we followed the guidelines in prior works [Ulmer et al., 2020, Azizmalayeri et al., 2023]. More specifically, we employed the pipeline provided in Sheikhalishahi et al. [2020][2] to keep patients with a length of stay of at least 48 hours, age greater than 18, and known discharge status. Since some of the variables are not available for some patients, they have suggested a list of more frequent variables provided in Table 4 to be used in the analysis. Patients lacking data for any of these variables are excluded from the dataset, which resulted in 54826 unique patients. Moreover, they aggregate the time-dependent variables through 6 different statistics including minimum, maximum, mean, standard deviation, skewness, and number of observations computed over windows consisting of the full time-series and its first and last 10%, 25%, and 50%. This dataset is then used for the mortality prediction task, where the data collected in the first 48 hours from patients is used to predict in-hospital mortality. The mortality rate in this dataset is 6.77%.

**MIMIC-IV:** The Medical Information Mart for Intensive Care (MIMIC) dataset provided critical care data for patients

---

| Description | eICU | MIMIC-IV |
|---|---|---|
| • *Time-dependent* | | |
| Blood pH value | pH | pH |
| Body temperature | Temperature (c) | Temperature |
| Respiratory rate | Respiratory Rate | Respiratory rate |
| Blood oxygen saturation | O2 Saturation | Oxygen saturation |
| Mean arterial pressure | MAP (mmHg) | Mean blood pressure |
| Heart rate | Heart Rate | Heart Rate |
| Blood glucose level | glucose | Glucose |
| Glasgow coma scale (total) | GCS Total | - |
| Glasgow coma scale (motor functions) | Motor | - |
| Glasgow coma scale (eyes) | Eyes | - |
| Glasgow coma scale (verbal) | Verbal | - |
| Fraction of inspired oxygen | FiO2 | - |
| Diastolic Blood Pressure | Invasive BP Diastolic | Diastolic blood pressure |
| Systolic Blood Pressure | Invasive BP Systolic | Systolic blood pressure |
| • *Time-independent* | | |
| Gender | gender | gender |
| Age | age | age |
| Ethnicity | ethnicity | - |
| Height at admission time | admissionheight | - |
| Weight at admission time | admissionweight | - |
| Admission type | - | admission_type |
| First care unit | - | first_careunit |

Table 4: Clinical variables used for each of eICU and MIMIC-IV datasets. Dash means that the variable is not included in the dataset.

admitted to ICU at the Beth Deaconess Medical Center. This dataset is accessible via credentializing on the PhysioNet website [3]. This dataset is pre-processed mainly similar to the eICU dataset. Initially, the data undergoes the pre-processing pipeline presented in Gupta et al. [2022][4], followed by filtering procedures similar to those employed in eICU. This resulted in 18180 unique patients in this dataset. This dataset is used for mortality prediction as in eICU and has a mortality rate of 12.57%.

**Diabetic Retinopathy Debrecen:** Diabetic retinopathy is a kind of diabetes that affect eyes by by damaging the blood vessels of the light-sensitive tissue. To facilitate the research studies on diabetic retinopathy, Messidor [Decencière et al., 2014], a collection of Diabetic Retinopathy examinations consisting of two macula-centered eye fundus images, is provided. Diabetic Retinopathy Debrecen is a dataset containing features extracted from the Messidor images to predict whether an image has signs of diabetic retinopathy. This dataset contains 1151 instances and can be accessed through the UCI machine learning repository [5]. This dataset is a balanced data and the proportion of positive labels is 53.1%.

**Dry Bean:** This dataset contains features from 7 different registered varieties of dry beans. It can be accessed through the UCI machine learning repository [6] and contains 13611 instances. The task in this dataset involves distinguishing various types of dry beans.

**Wine Quality:** This dataset combines data from the red and white vinho verde wine samples from the north of Portugal. It is publicly available through the UCI machine learning repository [7] and contains 4898 instances in total. This dataset is designed to predict the quality of wine based on physicochemical tests but can be used for color classification as well. In this study, we leverage it for the latter purpose.

---

[3] https://physionet.org/content/mimiciv/2.2/
[4] https://github.com/healthylaife/MIMIC-IV-Data-Pipeline
[5] https://archive.ics.uci.edu/dataset/329/diabetic+retinopathy+debrecen
[6] https://archive.ics.uci.edu/dataset/602/dry+bean+dataset
[7] https://archive.ics.uci.edu/dataset/186/wine+quality

## B.2 IMAGES

**MNIST:** The MNIST dataset is a collection of handwritten digits. The images in this dataset are grayscale and have a shape of 28×28. It has a training and test set including 60,000 and 10,000 instances, respectively. The prediction task in this dataset involves classifying the digits.

**Fashion-MNIST:** This dataset contains images from Zalando's articles, consisting of 60,000 images in the training set and 10,000 in the test set. The images size are 28×28.

**SVHN:** This dataset contains 600,000 images of digits used in house numbers. The images are 32×32 and colored.

**CIFAR-10:** The CIFAR-10 dataset consists of 10 classes of colored 32×32 images. The training and test contain 50,000 and 10,000 instances, respectively.

**CIFAR-100:** The CIFAR-100 dataset consists of 100 classes of colored 32×32 images. Each class in the training and test set contains 500 and 100 images, respectively, which makes it as large as the CIFAR-10 dataset.

## C   PROOF OF THEOREM 1

Before we come to Theorem 1, we first present a lemma needed in the proof.

**Lemma 1.** *Let $x \in R^D$ such that, for a given class $c$ in the output of the softmax function $\sigma$, the following limit holds:*

$$\lim_{x_d \to \infty} \sigma(f(x))_c = 1, \tag{8}$$

*where $f$ denotes an arbitrary function. Then, we can infer that:*

$$\exists\, c', \lim_{x_d \to \infty} f(x)_{c'} = \infty. \tag{9}$$

*Proof.* Given the continuity of the softmax, we can rewrite the limit of the composition of the two functions as

$$\sigma[\lim_{x_d \to \infty} f(x)]_c = 1, \tag{10}$$

Unfolding the definition of the softmax and moving the denominator across the equality we can conclude that

$$e^{\lim_{x_d \to \infty} [f(x)]_c} = \sum_{c'=1}^{|C|} e^{\lim_{x_d \to \infty} [f(x)]_{c'}}, \tag{11}$$

Since the output of the exponential is always a positive number larger than 0, the previous equation cannot hold if for all class indexes $c'$, $\lim_{x_d \to \infty} [f(x)]_{c'}$ is a finite number. Hence, for at least one index said limit must equal (plus or minus) infinity.

$\square$

**Theorem 1.** *Let $x \in R^D$ and suppose $\alpha$ is a scaling vector. Now $x' = \alpha \odot x$ can be considered as an OOD example if $\alpha$ is large enough. Let $h_\theta$ be any neural network whose last layer is linear, genearting an overconfident prediction for class $c$ on $x'$ in a C-class classification as:*

$$\lim_{\alpha_d \to \infty} \sigma(h_\theta(x'))_c = 1, \tag{12}$$

*where $\sigma$ is the Softmax function. Then, we infer that there exists at least a dimension in which the output of the penultimate layer goes to infinity in the limit:*

$$\exists\, k, \lim_{\alpha_d \to \infty} (x'_{R-1})_k = \infty, \tag{13}$$

*where $x'_{R-1} \in R^{D'}$ is the output of the penultimate layer.*

*Proof.* Let $w_R \in R^{C \times D'}$ and $b_R \in R^C$ denote the weights and biases at the last linear layer of the neural network $h_\theta$. Then, $h_\theta(x')$ can be formulated as:

$$h_\theta(x') = w_R x'_{R-1} + b_R. \tag{14}$$

Furthermore, by Lemma 1, Equation 12 implies that:

$$\exists\, c', \lim_{\alpha_d \to \infty} h_\theta(x')_{c'} = \infty. \tag{15}$$

Now, substituting Equation 14 into Equation 15, we have:

$$\exists\, c', \lim_{\alpha_d \to \infty} (w_R x'_{R-1} + b_R)_{c'} = \infty, \tag{16}$$

where $b_R$ is just a vector of scalar values, which can be disregarded in this limit. So, we rewrite the matrix multiplication for index $c'$ in Equation 16 as:

$$\exists\, c', \lim_{\alpha_d \to \infty} \sum_{k=1}^{D'} (w_R)_{c',k} \, (x'_{R-1})_k = \infty, \tag{17}$$

where $D'$ is the set of indices of the output of the penultimate layer. This entails that at least one of the members of the sum must tend to infinity. Since $(w_R)_{c',k}$ is just a scalar value, we deduce that:

$$\exists\, k, \lim_{\alpha_d \to \infty} (x'_{R-1})_k = \infty. \tag{18}$$

This means that the feature vector at the penultimate layer consists of at least one value that goes to infinity in the limit, completing the proof. $\qquad\square$

## D ADDITIONAL RESULTS

This section includes some extra results for the experiments discussed in the main text.

### D.1 TABULAR DATA

Results for three other tabular datasets are illustrated in Fig. 5, which aligns with the other tabular datasets discussed in the main text.

### D.2 ARCHITECTURES AND LOGITNORM

In the results presented for other types of architectures and LogitNorm training, we only included some of the baseline detection methods to keep the page limit. Results for other detection methods are displayed in tables 5 and 6. Conclusions on the baselines included in these tables are the same as the others discussed in the results section.

### D.3 EXTENSION TO IMAGES

Results for the CIFAR-100 dataset are displayed in Table 7. According to this table, when CIFAR-100 is the ID set, our method can improve results significantly within synthesized OOD sets, but not on real-world OOD sets. For instance, results on real-world OOD sets are the same with and without our method in the ResNet-32 model. This table also provides results for detection methods not included in the main text for the MNIST and CIFAR-10 datasets due to page limits, which follow the same trend as those in the main text.

## E MODEL CALIBRATION

Overconfidence and calibration on ID sets are not the same, since whether a model is calibrated on ID data may not influence how its confidence changes on OOD points, because by definition OOD instances come from a different distribution. Moreover, theoretical results demonstrate that some architectures are always overconfident, regardless of their level of calibration on ID data [Hein et al., 2019, Ulmer and Cinà, 2021].

This said, there may indeed be architectures for which an improved calibration enhances OOD detection, but as far as we know this remains to be proven in general. To assess the impact of calibration on our outcomes, we have included

Table 5: AUC of OOD detection with and without CEA using tabular ResNet and Transformer as the prediction model. We use eICU and Diabetics as ID and synthesize the OOD data by scaling factor $\alpha$. Superior results are emphasized in bold unless the two are equal. This table is similar to Table 1, but includes different baseline detection methods.

| ID | Method | ResNet | | | Transformer | | |
| | | $\alpha = 10$ | $\alpha = 100$ | $\alpha = 1000$ | $\alpha = 10$ | $\alpha = 100$ | $\alpha = 1000$ |
| | | Baseline / Baseline&CEA | | | | | |
|---|---|---|---|---|---|---|---|
| eICU | RMDS | 52.6 / **66.4** | 64.7 / **85.5** | 79.4 / **93.4** | 52.2 / **52.4** | 60.8 / **61.0** | 72.9 / **73.2** |
| | SHE | 72.9 / 72.9 | **89.9** / 89.8 | 93.8 / **93.9** | 57.7 / 57.7 | 73.2 / 73.2 | 81.5 / **81.6** |
| | KLM | 58.7 / **66.8** | 73.0 / **85.6** | 82.9 / **93.3** | 56.0 / **56.2** | 65.1 / **66.1** | 72.8 / **73.2** |
| | OpenMax | 58.6 / **69.2** | 69.6 / **87.1** | 79.6 / **93.6** | 51.1 / **51.7** | 54.2 / **56.1** | 71.2 / **72.6** |
| | MLS | 46.5 / **69.2** | 28.8 / **87.1** | 13.3 / **93.6** | 51.7 / **52.4** | 56.3 / **58.0** | 71.8 / **73.4** |
| | TempScale | 47.9 / **69.5** | 30.6 / **87.3** | 13.2 / **93.6** | 51.7 / **52.5** | 56.1 / **58.3** | 71.7 / **73.5** |
| | GradNorm | 37.1 / **66.5** | 17.3 / **84.9** | 7.2 / **93.4** | 53.5 / **54.1** | 63.1 / **64.2** | 76.3 / **77.1** |
| | DICE | 42.2 / **67.0** | 24.3 / **85.6** | 10.0 / **93.4** | 53.3 / **53.8** | 62.8 / **63.6** | 76.0 / **76.8** |
| | ASH | 47.2 / **69.5** | 30.8 / **87.2** | 14.4 / **93.5** | 51.7 / **52.4** | 56.0 / **57.8** | 70.5 / **72.1** |
| Diabetics | RMDS | 74.8 / **77.4** | 85.8 / **87.5** | 90.1 / **91.0** | 65.3 / **65.4** | 80.2 / **80.3** | 86.3 / 86.3 |
| | SHE | 81.9 / 81.9 | 88.7 / 88.7 | 91.5 / 91.5 | 69.2 / **69.3** | 85.3 / 85.3 | 90.0 / 90.0 |
| | KLM | 74.0 / **77.2** | 84.9 / **87.0** | 89.8 / **90.4** | 55.9 / **56.2** | 57.8 / **57.9** | 56.4 / **57.3** |
| | OpenMax | 55.8 / **73.7** | 66.2 / **86.6** | 68.8 / **90.2** | 42.7 / **43.9** | 50.0 / **50.6** | 64.2 / **65.6** |
| | MLS | 33.6 / **67.8** | 22.7 / **86.2** | 18.4 / **90.2** | 41.0 / **41.9** | 50.3 / **50.7** | 66.3 / **67.4** |
| | TempScale | 25.6 / **67.4** | 15.5 / **86.3** | 10.5 / **90.3** | 38.4 / **39.9** | 47.3 / **48.2** | 61.8 / **62.7** |
| | GradNorm | 21.4 / **65.2** | 13.2 / **85.7** | 9.7 / **90.0** | 38.7 / **39.7** | 47.6 / **48.2** | 62.2 / **63.0** |
| | DICE | 22.5 / **66.1** | 12.9 / **85.4** | 9.5 / **89.8** | 63.1 / **64.0** | 82.6 / **82.8** | 88.9 / **89.0** |
| | ASH | 35.0 / **68.9** | 23.3 / **86.1** | 17.8 / **90.2** | 43.1 / **44.0** | 51.5 / **52.0** | 65.2 / **65.6** |

temperature scaling [Guo et al., 2017] combined with MSP among our baselines (called TempScale). Table 8 presents the expected calibration error (ECE) with and without temperature scaling for the ResNet model trained on our datasets, which quantifies the improvement in ID calibration of models after Temp Scaling. Also, the OOD detection results stated before indicate that temperature scaling improves the AUC of OOD detection over MSP only marginally: at most 1% across all the experiments (e.g., see Figures 2 and 3). This suggests that ID calibration might be beneficial for OOD detection, although it is not enough to make such a claim.

# F  ALL LAYERS INSTEAD OF PENULTIMATE LAYER

In the proposed method, we utilize the activation values at the penultimate layer of the neural network. Here, we examine the impact of using all intermediate layers of neural networks rather than just one. To this end, we repeat the experiment from section 4.1 using $\alpha = 1000$ to consider all the layers. The same algorithm that was applied to the penultimate layer is now applied to all layers and the outputs (normalized by the number of nodes in their respective layers) are summed together. Fig. 6 displays the comparison between only one or all layers, for eICU and Diabetics datasets.

According to this figure, both settings are effective in improving the OOD detection performance. However, the performance of all the detection methods is better with only one layer with the eICU dataset, while with the Diabetics dataset, many baselines get better results when all the layers are employed. Accordingly, while both setups are effective, the best option depends on the dataset and detection method. Still, note that the average performance on these two datasets is better using only the penultimate layer.

# G  OTHER NORMS IN CEA

CEA measures the $\ell_2$ norm of activations exceeding a specified threshold. As stated in the main text, the choice of $\ell_2$ norm can potentially be substituted with other $\ell_p$ norms. Here, we evaluate how the utilization of $\ell_0$ and $\ell_1$ norms influences the outcomes. According to the results in Table 9, these norms result in similar results to the $\ell_2$ norm. This means that CEA can

Table 6: AUC of OOD detection with and without CEA on the model trained with the LogitNorm loss. We use eICU and Diabetics as ID and synthesize the OOD data by scaling factor $\alpha$. Superior results are emphasized in bold unless the two are equal. This table is similar to Table 2, but includes different baseline detection methods.

| ID | $\alpha$ | RMDS | SHE | KLM | OpenMax | MLS | TempScale | GradNorm | DICE | ASH |
|---|---|---|---|---|---|---|---|---|---|---|
| | | \multicolumn{9}{c}{Baseline / Baseline&CEA} | | | | | | | | |
| eICU | 10 | 61.3 / **64.9** | 65.3 / 65.3 | 51.5 / **63.9** | 52.6 / **65.7** | 55.1 / **67.3** | 55.1 / **67.3** | 33.9 / **60.5** | 38.6 / **63.0** | 57.3 / **65.4** |
| | 100 | 74.8 / **80.1** | 80.9 / 80.9 | 52.0 / **78.6** | 53.6 / **80.0** | 61.8 / **82.7** | 61.8 / **82.7** | 18.9 / **76.1** | 25.3 / **77.8** | 62.4 / **80.1** |
| | 1000 | 85.2 / **89.1** | 89.9 / **90.0** | 52.4 / **89.0** | 54.3 / **89.2** | 64.6 / **90.0** | 64.6 / **90.0** | 10.3 / **87.9** | 14.0 / **88.3** | 63.2 / **89.3** |
| Diabetics | 10 | 68.9 / **73.3** | 78.8 / 78.8 | 56.7 / **73.8** | 60.8 / **74.9** | 35.2 / **65.1** | 35.2 / **65.1** | 21.4 / **60.9** | 19.1 / **60.1** | 19.2 / **60.2** |
| | 100 | 84.2 / **86.8** | 88.4 / 88.4 | 59.3 / **86.0** | 63.8 / **86.7** | 32.6 / **85.7** | 32.6 / **85.7** | 12.6 / **84.6** | 11.5 / **83.9** | 11.6 / **84.1** |
| | 1000 | 89.7 / **90.8** | 91.5 / 91.5 | 62.1 / **90.0** | 65.6 / **90.6** | 31.8 / **90.1** | 31.8 / **90.1** | 9.5 / **89.6** | 8.7 / **89.0** | 8.8 / **89.1** |

be used with other reasonable norms as well.

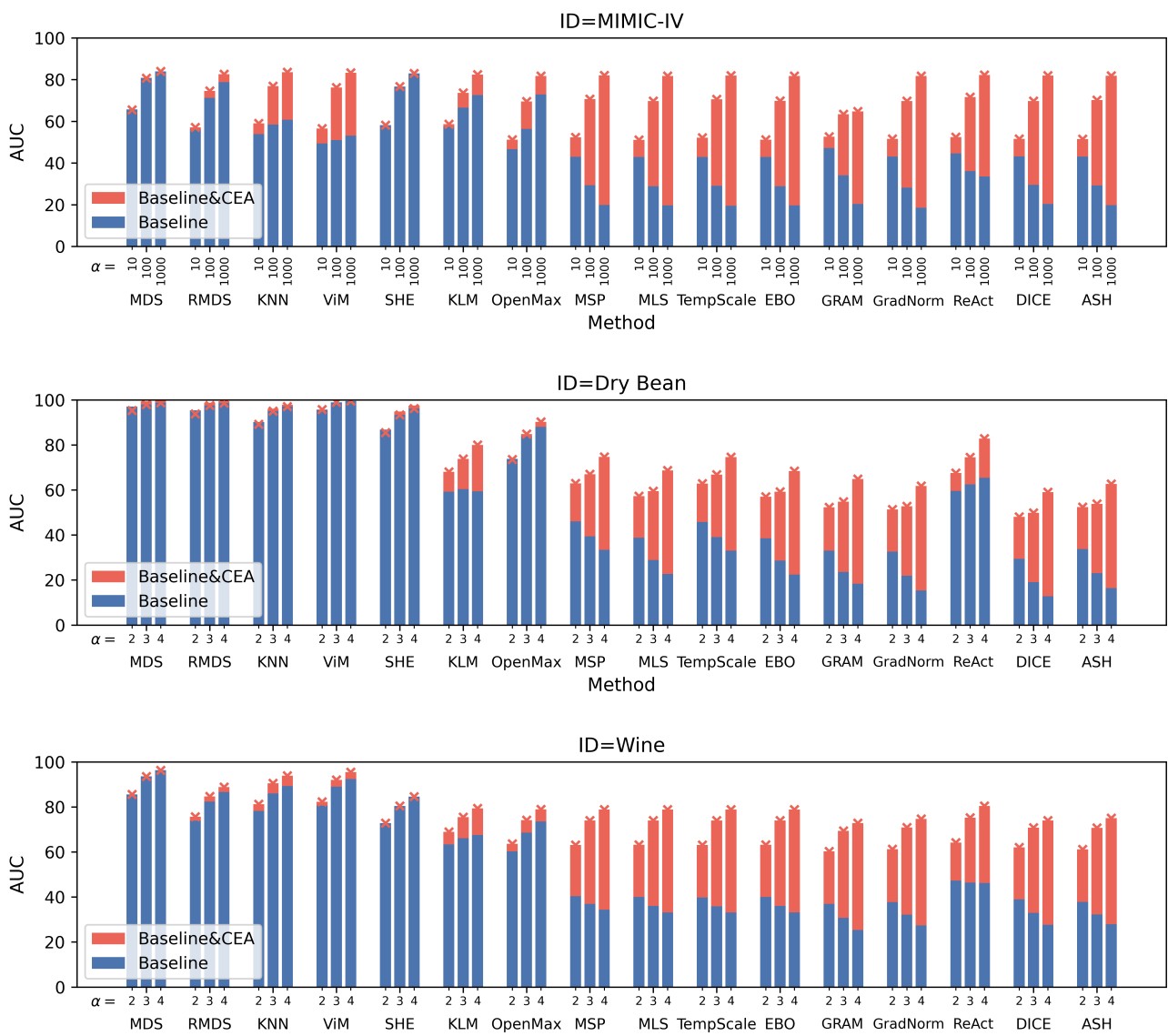

Figure 5: OOD detection performance with and without CEA using the MIMIC-IV (top), Dry Bean (middle), and Wine Quality (bottom) datasets as ID and synthesized data by scaling. The blue bars are positioned in front of the red ones and cross markers are employed to emphasize the top of the red bars. The scaling factors and baseline names are presented under each bar.

Table 7: AUC of OOD detection with and without CEA in image datasets. MNIST, CIFAR-10, and CIFAR-100 serve as ID, and OOD sets are synthesized by i) scaling or ii) an adversarial attack, or iii) selected from other datasets. ResNet-32 and ReLU MLP classifiers are used as the prediction model. Superior results are in bold unless the two are equal. This table is similar to Table 3, but includes CIFAR-100 as an ID dataset and different baseline detection methods.

| ID | Method | ReLU MLP | | | ResNet-32 | | |
|---|---|---|---|---|---|---|---|
| | | Scale | Attack | Other | Scale | Attack | Other |
| | | Baseline / Baseline&CEA | | | | | |
| MNIST | RMDS | 62.6 / **62.7** | 83.6 / **84.1** | 96.7 / **96.9** | 68.0 / 68.0 | 99.7 / 99.7 | 99.6 / 99.6 |
| | SHE | 63.3 / 63.3 | 92.1 / **93.2** | **86.3** / 85.9 | 61.0 / 61.0 | 99.9/99.9 | 99.7 / 99.7 |
| | KLM | 57.9 / **64.9** | 69.5 / **80.8** | 79.5 / **85.8** | 56.5 / **59.2** | 45.3 / **98.7** | 85.5 / **96.6** |
| | OpenMax | 62.7 / **62.8** | 85.8 / **88.5** | 90.4 / **92.0** | 57.2 / **57.4** | 99.2 / **99.6** | 99.4 / **99.5** |
| | MLS | 45.9 / **56.5** | 7.4 / **42.5** | 81.2 / **92.7** | 50.6 / **56.5** | 2.7 / **98.8** | 76.5 / **99.0** |
| | TempScale | 49.8 / **50.8** | 15.3 / **65.6** | 80.3 / **89.2** | 53.4 / **55.9** | 6.0 / **99.4** | 94.8 / **98.4** |
| | GradNorm | 43.4 / **57.2** | 5.9 / **41.3** | 39.0 / **61.6** | 37.4 / **64.4** | 1.3 / **98.9** | 62.8 / **98.3** |
| | DICE | 42.1 / **59.1** | 5.9 / **43.1** | 59.3 / **81.9** | 38.0 / **63.9** | 3.2 / **99.5** | 57.5 / **95.1** |
| | ASH | 42.2 / **60.1** | 7.5 / **42.5** | 81.2 / **92.7** | 51.4 / **55.3** | 2.7 / **98.8** | 76.1 / **98.9** |
| CIFAR-10 | RMDS | 89.8 / **95.4** | 53.6 / **68.7** | 58.5 / **61.3** | 98.0 / 98.0 | 99.9 / 99.9 | 87.4 / 87.4 |
| | SHE | 98.0 / 98.0 | **95.1** / 94.9 | 60.6 / 60.4 | 96.8 / 96.8 | 99.9 / 99.9 | 85.8 / 85.8 |
| | KLM | 88.8 / **96.1** | 90.9 / **93.9** | 57.7 / **59.0** | 78.9 / **94.1** | 68.4 / **92.2** | 80.5 / 80.5 |
| | OpenMax | 87.0 / **95.9** | 75.1 / **85.3** | 71.4 / **71.7** | 96.2 / **96.7** | 99.9 / 99.9 | 86.7 / **87.1** |
| | MLS | 20.2 / **94.0** | 8.4 / **35.5** | 67.7 / **72.5** | 64.3 / **97.6** | 0.0 / **83.8** | 90.1 / **90.2** |
| | TempScale | 19.3 / **92.2** | 9.6 / **53.9** | 58.2 / **64.2** | 93.1 / **98.5** | 0.0 / **99.6** | 88.3 / **88.9** |
| | GradNorm | 3.7 / **86.9** | 4.0 / **20.9** | 49.4 / **55.4** | 11.9 / **71.5** | 0.0 / **91.9** | 66.3 / **67.3** |
| | DICE | 4.8 / **92.5** | 4.8 / **55.1** | 58.9 / **68.0** | 52.4 / **93.1** | 0.0 / **88.6** | 90.1 / 90.1 |
| | ASH | 22.5 / **95.1** | 9.1 / **50.3** | 68.5 / **72.6** | 68.7 / **97.1** | 0.0 / **84.4** | 90.3 / 90.3 |
| CIFAR-100 | MDS | 96.7 / **97.0** | **94.5** / 94.4 | 57.7 / 57.7 | 98.7 / 98.7 | 99.9 / 99.9 | 53.1 / 53.1 |
| | KNN | 76.5 / **96.3** | 56.5 / **86.2** | 63.4 / **66.3** | 99.1 / 99.1 | 64.8 / **97.3** | 76.0 / 76.0 |
| | ViM | 65.2 / **96.1** | 72.3 / **79.2** | 60.0 / **61.6** | 5.6 / **95.2** | 0.1 / **83.7** | 82.1 / 82.1 |
| | MSP | 8.0 / **84.7** | 11.1 / **31.7** | 49.7 / **50.9** | 29.0 / **97.6** | 2.0 / **91.5** | 75.0 / **75.1** |
| | EBO | 55.0 / **94.7** | 64.6 / **75.6** | 60.2 / **61.6** | 5.6 / **95.2** | 0.1 / **83.3** | 82.1 / 82.1 |
| | ReAct | 89.0 / **95.4** | 86.7 / **89.8** | 59.7 / **61.7** | 93.2 / **98.4** | 12.8 / **91.7** | 77.7 / 77.7 |
| | Gram | 4.9 / **11.4** | 9.4 / **19.5** | 54.4 / **55.5** | 1.2 / **79.6** | 0.0 / **62.9** | 72.3 / 72.3 |
| | RMDS | 89.0 / **90.5** | 45.0 / **52.6** | 53.4 / **53.5** | 99.2 / 99.2 | 99.8 / **99.9** | 71.7 / **71.8** |
| | SHE | **97.6** / 97.4 | 94.3 / **94.5** | 58.0 / **58.6** | 98.5 / 98.5 | 99.9 / 99.9 | **57.1** / 56.9 |
| | KLM | 93.8 / **95.5** | 84.5 / **93.3** | 59.5 / **60.4** | 55.4 / **98.1** | 42.6 / **98.4** | 75.5 / 75.5 |
| | OpenMax | 78.5 / **82.3** | 79.9 / **86.0** | 55.1 / **55.9** | 62.7 / **84.7** | 60.3 / **93.5** | 70.0 / **70.1** |
| | MLS | 46.9 / **67.7** | 36.0 / **49.2** | 64.8 / **64.9** | 4.4 / **97.9** | 0.0 / **98.7** | 81.0 / **81.5** |
| | TempScale | 8.2 / **75.8** | 9.2 / **31.1** | 48.5 / **49.0** | 34.1 / **98.1** | 1.7 / **99.6** | 75.5 / **75.8** |
| | GradNorm | 3.4 / **78.3** | 5.2 / **14.4** | 46.7 / **47.6** | 1.5 / **96.7** | 0.0 / **99.5** | 80.9 / 80.9 |
| | DICE | 5.0 / **87.6** | 5.6 / **57.5** | 51.8 / **55.5** | 3.8 / **96.7** | 0.1 / **99.4** | 82.5 / 82.5 |
| | ASH | 56.8 / **93.6** | 55.4 / **76.8** | 62.1 / **62.8** | 2.9 / **98.4** | 0.0 / **98.9** | 82.1 / 82.1 |

Table 8: The ECE (%) using M = 15 bins with and without temperature scaling for the ResNet model trained on our datasets.

| Temp Scaling | MNIST | CIFAR10 | CIFAR100 | eICU | MIMIC-IV |
|---|---|---|---|---|---|
| ✗ | 0.38 | 4.47 | 13.33 | 1.81 | 5.43 |
| ✓ | 0.31 | 1.28 | 4.39 | 1.17 | 2.63 |

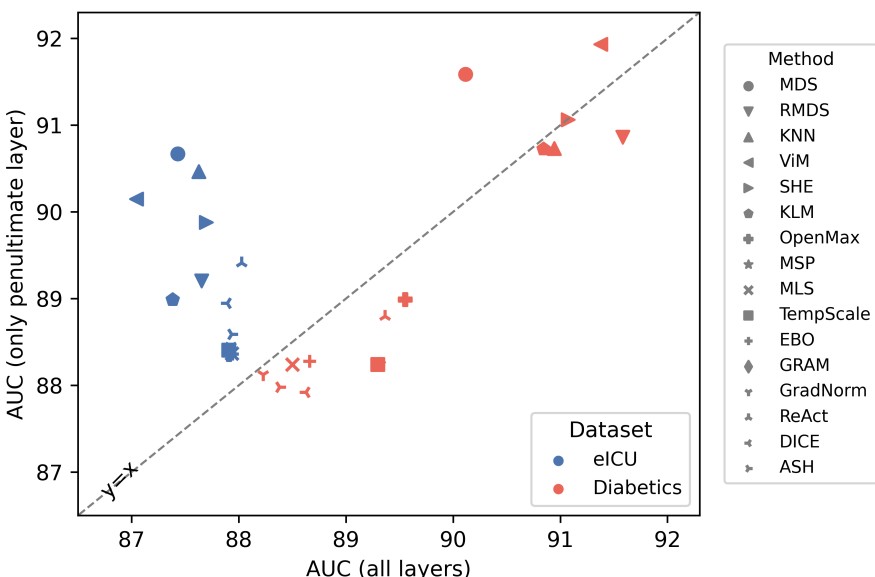

Figure 6: OOD detection performance with capturing extreme values in only the penultimate layer (y-axis) and in all the intermediate layers (x-axis). The eICU and Diabetics datasets serve as ID, and the OOD set is generated using $\alpha = 1000$.

Table 9: AUC of OOD detection with CEA using $\ell_0$, $\ell_1$, or $\ell_2$ norms to calculate size of extreme activations. Datasets include eICU and Diabetics, OODs are synthesized by $\alpha = 1000$, and baseline detection methods are MSP and EBO.

| Method | eICU | | | Diabetics | | |
|--------|------|------|------|-----------|------|------|
| | $\ell_0$ | $\ell_1$ | $\ell_2$ | $\ell_0$ | $\ell_1$ | $\ell_2$ |
| MSP | 88.2 | 88.3 | 88.4 | 88.1 | 88.2 | 88.2 |
| EBO | 88.4 | 88.4 | 88.4 | 88.2 | 88.2 | 88.3 |