# OpenReview forum: "Mitigating Overconfidence in Out-of-Distribution Detection by Capturing Extreme Activations"
_auai.org/UAI/2024/Conference — UAI 2024 poster_

### Official Review · Reviewer_Trt3 · 2024-03-16

**Q2-1 Originality-Novelty:** 2
**Q2-2 Correctness-Technical Quality:** 4
**Q2-5 Clarity Of Writing:** 3

**Q1 Summary And Contributions:**

This paper implements a practical post-hoc method (CEA) to address the problem of overconfidence in OOD detection. It uses the activations of the penultimate layer as indicators of overconfidence, and achieves promising empirical results when combined with diverse OOD methods.

**Q2-3 Extent To Which Claims Are Supported By Evidence:**

3: Good: the main claims are supported by convincing evidence (in the form of adequate experimental evaluation, proofs, (pseudo-)code, references, assumptions).

**Q2-4 Reproducibility:**

4: Excellent: key resources (e.g. proofs, code, data) are available and key details (e.g. proof sketches, experimental setup) are comprehensively described for competent researchers to confidently and easily reproduce the main results.

**Q3 Main Strengths:**

1. CEA is easy to use, and can be smoothly combined with other OOD detection methods.

2. Comprehensive experiments across diverse OOD detection methods support the effectiveness of CEA.

**Q4 Main Weakness:**

1. The motivation of CEA is somehow weak.

2. Some important comparisons are lacking.

**Q5 Detailed Comments To The Authors:**

1. The motivation of CEA seems to be from [Sun et al., 2021], which already has similar ideas on extreme activations. It is hard to evaluate how this paper improves over [Sun et al., 2021].

2. The choice of penultimate layer against other layers is not discussed in this paper. There should be at least references to the papers discussing it.

3. It is unclear which metric (maybe AUC?) is used in Table 1, 2, 3.

4. A kind suggestion: The authors may also consider other metrics like ECE.

**Q9 Complying With Reviewing Instructions:**

Yes

---

> ### Author Rebuttal · Authors · 2024-04-05
>
> We thank the reviewer for the valuable feedback. Here is our response to the specific comments, which overlaps with the weaknesses.
>
> 1. As stated in the related works, ReAct [Sun et al., 2021] caps the activation values at an upper limit to improve OOD detection. Despite the advantages of this method, we argue that this results in losing useful information. In contrast to ReAct, we suggest retaining those values when generating the novelty score, but adjusting the novelty score based on the $\ell_2$-norm of extreme activation values (CEA). In addition, we have also considered ReAct among the baseline OOD detection methods in the experiments. Results demonstrates that CEA is compatible with ReAct and can improve its performance in many scenarios. Accordingly, we believe that there is a clear improvement over ReAct both empirically and conceptually.
>
> 2. We have conducted a sensitivity analysis on the choice of penultimate layer in the Appendix E by considering all the layers instead of only the penultimate layer. Results demonstrate that using only the penultimate layer is better on average, but using all the layers can also be an option within some datasets and OOD detection methods.
>
>    Also, the choice of penultimate layer is standard in the literature [Lee et al. 2018, Zhang et al. 2022, Sun et al. 2022]. We have added the references to the text in section 3.2.
>
> 3. All the reported values are AUC. We have added this to the caption.
>
> 4. Overconfidence and calibration on ID sets are not the same, since whether a model is calibrated on ID data does not influence how its confidence changes on OOD points, because by definition OOD instances come from a different distribution. On the other hand, calibration on OOD sets is a matter of OOD generalization more than OOD detection. If a model generalizes well to all OOD sets then there is no real need for OOD detection. If we do OOD detection it is typically because we do not know what performance metrics to expect on OOD sets, thus calibration is information we often do not have at detection time. Case in point, in our experiments we often do not have labels for OOD datapoints, hence calibration cannot be measured on OOD data.
>
>     It is true that having excellent calibration - e.g. measured by ECE - on an OOD subset will prevent overconfidence in that subset (except for the corner case where labels also converge to a fixpoint). However, this does not necessarily generalize to other OOD subsets, so unless you have guarantees that you have good coverage on all OODs you still require some OOD detection mechanism.
>
>    So far we have discussed calibration on the main task. If the reviewer meant calibration on the OOD detection task, this is something we can easily calculate and display. We had not done it since we have never seen this metric reported in the OOD detection literature.

---

### Official Review · Reviewer_XND9 · 2024-03-17

**Q2-1 Originality-Novelty:** 3
**Q2-2 Correctness-Technical Quality:** 3
**Q2-5 Clarity Of Writing:** 3

**Q1 Summary And Contributions:**

This paper addresses the challenge of overconfidence in out-of-distribution (OOD) detection in neural networks. It introduces a novel approach that improves OOD detection by incorporating extreme activation values in the penultimate layer of neural networks into the novelty score calculation. This method is model-agnostic and demonstrates significant improvements in OOD detection performance in comprehensive experiments across multiple datasets, architectures, and scenarios.

**Q2-3 Extent To Which Claims Are Supported By Evidence:**

3: Good: the main claims are supported by convincing evidence (in the form of adequate experimental evaluation, proofs, (pseudo-)code, references, assumptions).

**Q2-4 Reproducibility:**

4: Excellent: key resources (e.g. proofs, code, data) are available and key details (e.g. proof sketches, experimental setup) are comprehensively described for competent researchers to confidently and easily reproduce the main results.

**Q3 Main Strengths:**

* The paper's method offers a model-agnostic, simple but novel solution to the overconfidence problem inspired by previous works.
* The authors do a comprehensive empirical evaluation across different datasets, architectures, and data types.
* The paper is well-structured clearly written and the method is explained well.

**Q4 Main Weakness:**

* The method depends on hyperparameter tuning for optimal performance, which might be a limitation, especially for users trying to apply the method out-of-the-box.
* A discussion of the method's limitations is missing.

**Q5 Detailed Comments To The Authors:**

* I recommend authors to explore hyperparameter sensitivity of the method and to offer guidelines for selecting hyperparameters.

* Further discuss other limitations of the approach. Are there any cases where data or model characteristics might affect the performance of the method?

**Q9 Complying With Reviewing Instructions:**

Yes

---

> ### Author Rebuttal · Authors · 2024-04-05
>
> We are grateful for the thoughtful review. Following are our responses to the comments provided.
>
> * In appendix D, we have conducted an ablation study on the hyperparameters, which demonstrates that the conclusions about CEA are robust within reasonable hyperparameter ranges. While we use a fixed set of hyperparameters within each modality, we agree that more hyperparameter tuning for optimal performance in each setting separately could improve our results even more.
>
>    As described in section 3.2, the main hyperparameters in our method are $\tau$ and $\lambda$, which are found based on a validation set from the in-distribution set and a fixed set of values for $p$ and $\gamma$. However, we could still tune $\tau$ and $\lambda$ further using a diverse set of OOD examples by checking the OOD detection performance within different ranges of these parameters. Note that this requires a set of OOD data which covers different kinds of OOD examples that model may face in practice, which is not available in many cases. This is why we have limited ourselves to finding the parameters based on a validation set from in-distribution data. We have extended the discussion on hyperparameter selection in section 3.2 to include these points.
>
> * We have revised the text to extend our reflections on the limitations in discussion section. As stated in paragraphs 3 to 5 of the discussion section, some OOD detection methods, prediction model architectures, and characteristics of data are inherently mitigating the overconfident issue, which reduces the added value of our method. Moreover, our study investigates the overconfidence issue in OOD detection empirically, but a theoretical study of the problem can also be a valuable contribution to the field. Finally, as discussed above, it may be not easy to find the optimal parameters without a set of diverse OOD examples.

---

### Official Review · Reviewer_GTu1 · 2024-03-22

**Q2-1 Originality-Novelty:** 2
**Q2-2 Correctness-Technical Quality:** 3
**Q2-5 Clarity Of Writing:** 4

**Q1 Summary And Contributions:**

The paper proposes a way to correct for overconfidence in OOD detectors. This takes the form of an extra term that can be added to existing OOD scores. The papers shows results on synthetic and real world datasets.

**Q2-3 Extent To Which Claims Are Supported By Evidence:**

3: Good: the main claims are supported by convincing evidence (in the form of adequate experimental evaluation, proofs, (pseudo-)code, references, assumptions).

**Q2-4 Reproducibility:**

2: Fair: key resources (e.g. proofs, code, data) are unavailable but key details (e.g. proof sketches, experimental setup) are sufficiently well-described for an expert to confidently reproduce the main results.

**Q3 Main Strengths:**

* The proposed method is simple and gives good results. In particular, it is never detrimental.
* The article is written very well.

**Q4 Main Weakness:**

* Most experiments use synthetic data. The OOD data is generated by scaling the in domain data by a large factor, [10, 100, 1000]. This results in samples that the proposed score function is very good at picking out. But these may not be representative of actual real world OOD data. The only experiment with real data is in table 3.

* The method is similar to MDS, which can also be seen from the results (combining the CEA score with MDS makes no difference).
  The pros and cons of CEA compared to MDS should be discussed.

* The possible variations of the method are not investigated.

**Q5 Detailed Comments To The Authors:**

* "github.com/anonymized-for-submission."
  You can use https://anonymous.4open.science/ to anonymize github links in paper submissions

* For the adversarial attack images: what is the loss function that is being attacked? Are these images adversarially maximizing the standard cross-entropy classification loss? Or are they optimizing some OOD score?

 * "The value returned by g(x) should monotonically increase as the overconfidence level rises, e.g., when amplifying α for synthesizing the OOD instances."
   What is alpha? Is this referring to the construction at the end of section 2? It is not immediately clear that this is the model that will be used throughout the paper. And later on in the experiment section, α is used slightly differently as well.

 * "we can measure the magnitude of CEA as a proxy for g(x)."
   What is CEA exactly?

 * "we define g(x) as: (3)"
   but the equation defines CEA(x), not g(x)

* "Table 1: OOD detection performance" what are these numbers? Is this AUC? If so, put it in the caption.

**Q9 Complying With Reviewing Instructions:**

Yes

---

> ### Author Rebuttal · Authors · 2024-04-05
>
> We thank the reviewer for the valuable feedback. Below is our response to each comment.
>
> **Q4**
>
> * Two series of real-world experiments are included in our study. The first one is on tabular datasets (Fig. 3), where we use eICU or MIMIC-IV as in-distribution set. These datasets are widely used in the medical domain with more than 6k citations. The second one is on image datasets (Table 3 and also Table 7 in the appendix), where we use MNIST, CIFAR-10, and CIFAR-100.
>
>    The benefit of synthetic data is that they allow for more control when studying the overconfidence issue, in line with previous theoretical works [Ulmer and Cina 2021]. Still, we agree that our experiments with real-world data give a better understanding of performance in the real-world.
>
> * We have revised the text to extend our discussion on this point in the third paragraph of section 5.  Methods like MDS address overconfidence inherently and CEA does not improve much the performance in such methods. However, it should be noted that that one cannot solely rely on such methods as they may not perform well in general, e.g. see ResNet-32 model trained on the CIFAR-10 dataset in Table 3. More specifically, another baseline combined with CEA may perform better than methods like MDS. Hence, CEA allows us to replace methods like MDS with other baselines while keeping the benefits of those methods.
>
>    To clarify, assume a case where MDS is not as good as another OOD detection method X. In this case, using MDS score alone or adding it to the novelty score of X would result in a lower OOD detection performance than using the original novelty score of X. In contrast, CEA is designed such that combining it with any detection method would not result in lower performance, while it can give a net positive improvement in many cases.
>
>
>
> * We have included some variations of the method in the appendix. In Appendix D, we conduct an ablation study on the hyperparameters of our method, which demonstrates that the conclusions about CEA are robust within reasonable hyperparameter ranges. Also, in Appendix E, we consider a setting where we use outputs of all the layers of neural network instead of only the penultimate layer. This experiment highlights that using only the penultimate layer is better on average, but using all the layers can also be an option in some cases. In addition, we have added a new appendix F where we report experiments with other norms instead of $\ell_2$ (namely $\ell_0$ and $\ell_1$) and show that results are essentially the same.
>
> **Q5**
>
> * Thanks for the suggestion, we will use it for future submissions. We have also put the code in the supplementary material, which is the same code as we are going to release on github.
>
> * We have maximized the standard cross-entropy classification loss. Our goal here is not to reduce the performance of OOD detection method, we just want to check the detection performance under an attack to the classification task. Moreover, the prediction model is not trained with adversarial training; hence, attacking the CE loss can also give examples that reduces the OOD detection performance. This is why we have not used an end-to-end attack to the OOD score.
>
> * We have revised the text to incorporate the reviewer’s points and improve clarity as follows:
>   * Alpha is the scaling factor used for constructing the synthetic data defined at the end of section 2. We have uniformed the presentation to clarify this.
>   *	We meant “We can measure the magnitude of extreme activations denoted as CEA (capturing extreme activations) as a proxy for g(x).”
>   *	It is the definition of CEA as the reviewer mentioned.
>   *	All the reported values are AUC. We have added this to the caption.

---

### Official Review · Reviewer_43Zp · 2024-03-27

**Q2-1 Originality-Novelty:** 3
**Q2-2 Correctness-Technical Quality:** 3
**Q2-5 Clarity Of Writing:** 3

**Q10 Ethical Concerns:**

not include

**Q1 Summary And Contributions:**

The paper presents a method to mitigate the overconfidence issue in neural network predictions for out-of-distribution (OOD) samples, which poses a problem for OOD detection. By measuring extreme activation values in the penultimate layer, the proposed method adjusts the novelty score used to detect OOD instances. This approach not only significantly improves OOD detection performance across a variety of datasets and architectures, including ResNet and Transformer, but also maintains in-distribution accuracy. The method is designed to be easily integrated with existing detection techniques, enhancing their reliability without substantial modifications, and the authors provide open-source code to ensure reproducibility.

**Q2-3 Extent To Which Claims Are Supported By Evidence:**

2: Fair: the main claims are somewhat supported by evidence (but the experimental evaluation may be weak, or does not match entirely with the claims, important baselines may be missing, proofs contain important ideas but lack rigor, algorithmic details are only discussed superficially, references are imprecise, assumptions are not sufficiently motivated or explicated, etc.).

**Q2-4 Reproducibility:**

3: Good: key resources (e.g. proofs, code, data) are available and key details (e.g. proofs, experimental setup) are sufficiently well-described for competent researchers to confidently reproduce the main results.

**Q3 Main Strengths:**

1. Innovative Approach: The paper introduces a novel method that incorporates extreme activation values to identify overconfidence in model predictions. This provides a new angle to tackle the problem of OOD detection that existing methods may overlook.
2. Comprehensive Evaluation: The method is evaluated across a diverse set of experiments that include different data types (synthetic and real-world), data formats (tabular and image), and neural network architectures (ResNet and Transformer). This thorough testing underscores the robustness and generalizability of the approach.
3. Significant Performance Improvements: The proposed method demonstrates substantial improvements in OOD detection performance over several baseline methods. The reported double-digit increases in AUC are indicative of its effectiveness.

**Q4 Main Weakness:**

Lack of Theoretical Analysis: The paper might lack a comprehensive theoretical backing for why capturing extreme activation values correlates with overconfidence. A more rigorous theoretical foundation could strengthen the claims.

**Q5 Detailed Comments To The Authors:**

1. Theoretical Justification: Expanding on the theoretical rationale behind why extreme activation values are indicative of overconfidence would strengthen the foundation of the method. Can the authors provide a more detailed explanation or proof that links these activations to overconfidence in a broader range of neural network architectures?
2. Broader Implications: The paper could elaborate on the broader implications of improved OOD detection, such as increased safety in AI deployment, particularly in critical applications like autonomous vehicles or healthcare.

**Q9 Complying With Reviewing Instructions:**

Yes

---

> ### Author Rebuttal · Authors · 2024-04-05
>
> We thank the reviewer for the helpful comments. Here is our detailed response.
>
> **Q4**
>
> This work is more a practical study, but we can offer the following theorem to support our intuitions and method. In a nutshell, it shows that overconfident behavior on the side of the model entails the presence of extreme activations in the penultimate layer. This entails that, in the presence of overconfidence, OOD detection methods such as MSP and entropy will (tend to) fail to detect extreme OOD examples, while CEA will be able to label them as OOD.
>
> *Theorem 1*. Let $x\in R^D$ and suppose $\alpha$ is a scaling vector. Now $x'= α\odot x$ can be considered as an OOD example if $\alpha$ is large enough. Let $h_θ$ be any neural network that generates an overconfident prediction for class c on $x'$ as:
>
> $
> \lim_{{\alpha_d \to \infty}} \sigma(h_\theta (x'))_c = 1,\quad (1)
> $
>
> where $\sigma$  is the Softmax function. Then, there is at least one value at the output of the penultimate layer that tends to infinity.
>
>
> *Proof*.
>
> The failure of MSP and entropy arises from their definitions, as they both return the lowest novelty score under equation 1. Hence, we just need to prove that the output of the penultimate layer consists of extreme values.
>
> Suppose $x_{R-1}'$ is the output of the penultimate layer and $w_R$  and $b_R$ are the weights and biases at the last linear layer of the neural network $h_θ$. Then, $h_θ(x')$ can be formulated as:
>
> $
> h_\theta (x') = w_R x_{R-1}' + b_R.\quad (2)
> $
>
> Also, from the definition of softmax, if the output of the softmax function tends to 1, it can be proven that at least one dimension in the input should go to infinity. Hence, from equation 1 we can conclude that
>
> $
> \exists\ c' \, \lim_{\alpha_d \to \infty} h_\theta (x')_{c'} \to \infty.\quad (3)
> $
>
> Now, by employing equation 2 in equation 3, we can conclude that there exist a dimension k in the vector $x_{R-1}'$ that tends to infinity in the limit:
>
> $
> \exists\ k \, \lim_{\alpha_d \to \infty} x_{R-1,\ k}' \to \infty.\quad (4)
> $
>
> This means that the feature vector at the penultimate layer consist of at least one value that goes to infinity in the limit. Therefore, if we capture the activation values in the penultimate layer that are larger than a threshold $\tau$ as we do in CEA, we can detect $x'$ as OOD.
>
>
> We have modified section 3.2 to include the statement of this theorem as the justification for our approach. The proof has been pushed in a new section in the appendix.
>
>
> **Q5**
>
> Improved OOD detection can enhance reliability of machine learning models when deployed in real-time. As the reviewer stated, this can benefit a wide range of domains that currently use machine learning models such as healthcare (e.g., disease recognition or mortality prediction), financial services (e.g., OOD patterns or fraud detection), transportation (e.g., autonomous vehicles), and Cybersecurity (e.g., identification of OOD network patterns). We have revised the Discussion to elaborate on the implications in more details.

---

### Official Review · Reviewer_YhKE · 2024-03-29

**Q2-1 Originality-Novelty:** 2
**Q2-2 Correctness-Technical Quality:** 3
**Q2-5 Clarity Of Writing:** 3

**Q10 Ethical Concerns:**

This paper does not raise any potential ethical concerns to the best of my understanding.

**Q1 Summary And Contributions:**

The paper aims at enhancing available methods of Out-of-Distribution (OoD) inputs detection utilizing the notorious overconfidence in their predictions. To that end, it considers extreme levels of activations at the penultimate layer in terms of l2-norm as overconfident. It encapsulates this logic in a separate threshold-based term added with a particular weight to the available OoD detection baseline methods. There are two hyperparameters to identify before applying the suggested algorithm: a threshold for the overconfident predictions and a weight for this additional term responsible for signaling if there is indeed an overconfidence. The threshold is chosen based on the validation dataset available during training and fine-tuning the model, whereas the weight is taken as the weighted ratio of the sum of values of a particular OoD detection method to the sum of values of the suggested overconfidence enhancement, both sums are computed based on the same validation dataset. Several deep neural network (DNN) architectures, including Transformers and ResNet, were used for evaluation. The datasets represent diversity and cover both tabular data and images. Numerous experiments demonstrate that the suggested term responsible for overconfidence significantly improves detection accuracy in most of the considered cases.

**Q2-3 Extent To Which Claims Are Supported By Evidence:**

3: Good: the main claims are supported by convincing evidence (in the form of adequate experimental evaluation, proofs, (pseudo-)code, references, assumptions).

**Q2-4 Reproducibility:**

3: Good: key resources (e.g. proofs, code, data) are available and key details (e.g. proofs, experimental setup) are sufficiently well-described for competent researchers to confidently reproduce the main results.

**Q3 Main Strengths:**

1. To the best of my knowledge, it represents a novel method of incorporating an overconfidence issue into almost any currently available approach that aims at OoD detection.
2. A diverse and rich baseline used in experiments with clear evidence of the positive added value of the suggested method.
3. A good range of datasets, including tabular and images, were used to evaluate the method.
4. A solid implementation that is available for reproduction of the results.

**Q4 Main Weakness:**

1. The idea that overconfidence is only inherent to the OoD inputs is somewhat flawed since it can be the case that a new instance appears in the wild that will be both overconfident and in-distribution. The paper does not claim that it equates OoD and overconfidence. However, it delegates this part to the available methods of OoD detection that should somehow tackle this issue themselves, which represents a significant weakness of the suggested method in theory.
2. There is nothing mentioned about the degree of calibration used during the evaluations. Since it represents an essential part of the overconfidence issue, this part constitutes another great weakness of the paper.
3. Finally, it is clear that the paper's main idea was inspired by the approach leveraged in [1]; it would be beneficial to consider another pole of the problem, namely, underconfidence of the predicted values: when are these related to the OoD? Can it be used to improve currently available OoD methods, etc.?

[1] Sun, Y., Guo, C., & Li, Y. (2021). ReAct: Out-of-distribution detection with rectified activations. In Advances in Neural Information Processing Systems (Vol. 34, pp. 144-157).

**Q5 Detailed Comments To The Authors:**

1. You claim that "the value returned by g(x) for ID data should be smaller or equal to the value returned by g(x) for OOD data" and also that "To this end, we use the ℓ2 norm of node activation values at the penultimate layer of the neural network that are higher than a specified threshold". I would be somewhat careful with such a claim. How could you guarantee that the threshold computed based on the validation dataset available during training and fine-tuning of the model would be valid for the instances that the model can meet in the wild?
2. What level of calibration for DNNs is used in your experiments since it is closely related to the issue you would like to address, i.e., overconfidence?
3. It seems that methods that already have a built-in tackling of overconfidence (e.g., MDS) are already good enough and cannot be improved with your method. What is an added value in these cases?
4. What about the other side of the spectrum: underconfidence? Have you considered evaluating and tackling these cases?

**Q9 Complying With Reviewing Instructions:**

Yes

---

> ### Author Rebuttal · Authors · 2024-04-05
>
> We thank the reviewer for the insightful remarks. Here is our response to the specific comments.
>
> **Q4**
> 1. It is true that model can be overconfident on some examples from in-distribution (ID); however, note that we find the threshold for capturing extreme activations based on a ID validation set such that almost all ID activations remain below the cutoff. Therefore, it is unlikely to encounter an instance sampled from the ID with activation values significantly exceeding the threshold. Moreover, this becomes even more unlikely when we have a large validation set that contains a diverse set of examples. Hence, if we have a large enough dataset, this would not be a significant issue for our method.
>
>    In addition, even if we assume that there are rare cases that are not present in the ID data (e.g., when a new instance appears in the wild), it is likely that the prediction model would be unreliable on these cases as they are not included in the ID data. Hence, it may be better to label them as OOD to avoid unreliable predictions.
>
> 2. Overconfidence and calibration on ID sets are not the same, since whether a model is calibrated on ID data does not influence how its confidence changes on OOD points, because by definition OOD instances come from a different distribution. On the other hand, calibration on OOD sets is a matter of OOD generalization more than OOD detection. If a model generalizes well to all OOD sets then there is no real need for OOD detection. If we do OOD detection it is typically because we do not know what performance metrics to expect on OOD sets, thus calibration is information we often do not have at detection time. Case in point, in our experiments we often do not have labels for OOD datapoints, hence calibration cannot be measured on OOD data.
>
>    It is true that having excellent calibration - e.g. measured by ECE - on an OOD subset will prevent overconfidence in that subset (except for the corner case where labels also converge to a fixpoint). However, this does not necessarily generalize to other OOD subsets, so unless you have guarantees that you have good coverage on all OODs you still require some OOD detection mechanism.
>
>    So far we have discussed calibration on the main task. If the reviewer meant calibration on the OOD detection task, this is something we can easily calculate and display. We had not done it since we have never seen this metric reported in the OOD detection literature.
>
> 3. Based on the definition provided for overconfidence (namely that confidence tends to one as the input is scaled), underconfidence can be interpreted as the model returning a flat probability for an input, which is already addressed in the existing OOD detection methods, denoted by f(x) in equation 1. One of the ideas behind some OOD detection methods is that the model will have underconfidence when it encounters OOD instances. For example, Maximum Softmax Probability (MSP) method expects a lower probability for the predicted class on the OOD examples compared to the examples from ID data. Therefore, underconfidence would not be an issue for such methods. In addition, as far as we know, no prior study has observed a detrimental effect of underconfidence on OOD detection.
>
> **Q5**
> 1. See answer to the point 1 in Q4. We have revised the text to include this point in the Discussion section.
>
> 2. See answer to the point 2 in Q4.
>
> 3. As discussed in third paragraph of section 5, methods like MDS address overconfidence inherently and CEA does not improve much the performance in such methods. However, it should be noted that that one cannot solely rely on such methods as they may not perform well in general, e.g. see ResNet-32 model trained on the CIFAR-10 dataset in Table 3. Therefore, another baseline combined with CEA may perform better than methods like MDS. Hence, CEA allows us to replace methods like MDS with other baselines while keeping the benefits of those methods.
>
> 4. See answer to the point 3 in Q4.

---

### Meta-Review · Area_Chair_Qg4W · 2024-04-11

This paper addresses the challenge of overconfidence in out-of-distribution (OOD) detection in neural networks. It introduces a novel approach that improves OOD detection by incorporating extreme activation values in the penultimate layer of neural networks into the novelty score calculation. This method is model-agnostic and demonstrates significant improvements in OOD detection performance in comprehensive experiments across multiple datasets, architectures, and scenarios.

The vision and novelty are clearly above the bar of UAI. For example, the paper introduces a novel method that incorporates extreme activation values to identify overconfidence in model predictions. This provides a new angle to tackle the problem of OOD detection that existing methods may overlook. Moreover, the method is evaluated across a diverse set of experiments that include different data types (synthetic and real-world), data formats (tabular and image), and neural network architectures (ResNet and Transformer). This thorough testing underscores the robustness and generalizability of the approach. More importantly, the proposed method demonstrates substantial improvements in OOD detection performance over several baseline methods. The reported double-digit increases in AUC are indicative of its effectiveness. While the reviewers had some concerns on the theoretical analysis, the authors did a particularly good job in their rebuttal. Please include the additional analysis and discussion in the next version.